# Leveraging Uncertainty Estimates to Improve Classifier Performance

**Gundeep Arora, Srujana Merugu, Anoop Saladi, Rajeev Rastogi**
Amazon
gundeepa@amazon.com

## Abstract

Binary classification involves predicting the label of an instance based on whether the model score for the positive class exceeds a threshold chosen as per application needs (e.g., maximizing recall at a precision bound). However, model scores are often not aligned with the true conditional probability of the positive class. This is especially true when the training involves differential sampling across classes or there is distributional drift between train and test settings. In this paper, we provide theoretical analysis and empirical evidence of the dependence of model score estimation bias on both uncertainty and score. Further, we formulate the decision boundary selection in terms of both model score and uncertainty, prove that it is NP-hard, and present algorithms based on dynamic programming and isotonic regression. Evaluation of the proposed algorithms on three real-world datasets yield 25%-40% gain in recall at high precision bounds over the traditional approach of using model score alone, highlighting the benefits of leveraging uncertainty.

## 1 Introduction

Many real-world applications such as fraud detection and medical diagnosis can be framed as binary classification problems, with the positive class instances corresponding to fraudulent cases and disease prevalence, respectively. When the predicted labels from the classification models are used to drive strict actions, e.g., blocking fraudulent orders and risky treatments, it is critical to minimize the impact of erroneous predictions. This warrants careful selection of the class decision boundary using the model output while managing the precision-recall trade-off as per application needs.

Typically, one learns a classification model from a training dataset. The class posterior distribution from the model is then used to obtain the precision-recall (PR) curve on a hold-out dataset with distribution similar to the deployment setting. Depending on the application need, e.g., maximizing recall subject to a precision bound, a suitable operating point on the PR curve is identified to construct the decision boundary. The calibration on the hold-out set is especially important for applications with severe class imbalance, since it is a common practice to downsample the majority[1] class during model training. This approach of downsampling followed by calibration on hold-out set is known to both improve model accuracy and reduce computational effort (Arjovsky et al., 2022).

A key limitation of the above widely used approach is that the decision boundary is based solely on the classification model score and does not account for the prediction uncertainty, which has been the subject of active research (Zhou et al., 2022; Sensoy et al., 2018). *A natural question that emerges is whether two regions with similar scores but different uncertainty estimates should be treated identically when constructing the decision boundary.* Recent work points to potential benefits of combining model score with estimates of uncertainty (Kendall & Gal, 2017) for specialized settings (Dolezal et al., 2022) or via heuristic approaches (Pocevivciūtė et al., 2022). However, there does not exist an in-depth analysis on why incorporating uncertainty leads to better classification, and how it can be adapted to any generic model in a post-hoc setting.

In this paper, we focus on binary classification with emphasis on the case where class imbalance requires differential sampling during training. For brevity, we refer to the conditional prob-

---

[1]Without loss of generality, we assume that the downsampling is performed on the -ve class (label=0) and the model score refers to +ve class (label=1) probability.

ability of the positive class as the *positivity rate*. Cognizant of the differences between the underlying distribution, train, and test sets, we refer to the corresponding positive class probabilities as *true, train and test positivity* respectively. We investigate four questions: **RQ1**: Does model score estimation bias (deviation from test positivity) depend on uncertainty? **RQ2**: If so, how can we construct an optimal 2D decision boundary using both model score and uncertainty and what is their relative efficacy? **RQ3**: Under what settings of undersampling of negative class and precision range do we gain the most in recall from incorporating uncertainty? **RQ4**: Do uncertainty estimates also aid in better calibration of class probabilities?

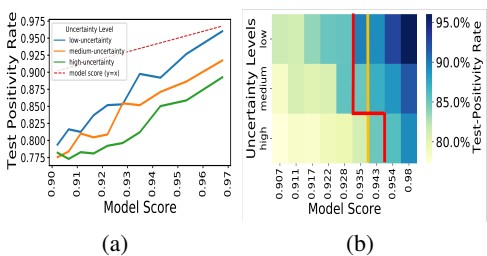

(a)                                (b)

Figure 1: (a) Test positivity rate vs. model score for different uncertainty levels on `Criteo` with 33% undersampling of negatives during training. (b) Heatmap of test positivity for different score and uncertainty ranges. Proposed method(red) yields better recall over vanilla score-based threshold (yellow).

Intuitively, choosing the decision boundary based on test positivity rate is likely to yield the best performance. However, the test positivity rate is not available beforehand and tends to differ from the model score as shown in Fig. 1(a). Our examination of **RQ1** indicates that the score estimation bias, i.e., difference between test positivity rate and the model score often varies with uncertainty in a systemic fashion. Specifically, for a representative setting with Beta priors, using Bayes rule, we observe that for input regions with a certain empirical train positivity rate, the "true positivity" (and hence test positivity rate) is shifted towards the global prior, with the shift being stronger for regions with low evidence. While Bayesian models try to adjust for this effect by combining the evidence, i.e., the observed train positivity with "model priors", there is still a significant bias when there is a mismatch between the model priors and true prior in regions of weak evidence (high uncertainty). Differential sampling across classes during training further contributes to this bias. This finding that the same model score can map to different test positivity rates based on uncertainty levels indicates that the decision boundary chosen using score alone is likely to be suboptimal relative to the one optimized using uncertainty and model score. Fig. 1(b) depicts maximum recall boundaries for a specified precision bound using score alone (yellow) and with both score and uncertainty estimates (red) validating this observation, which motivates the subsequent questions **RQ2-RQ4**.

**Contributions.** Below we summarize our contributions on leveraging the relationship between score estimation bias and uncertainty to improve classifier performance.

1. To motivate the need for incorporating uncertainty into decision making, we consider a representative Bayesian setting with Beta priors and Posterior Network (Charpentier et al., 2020) as an exemplary uncertainty estimation method, and demonstrate that the test positivity rate depends on both score and uncertainty, and monotonically increases with score for a fixed uncertainty. There is also a dependence on the downsampling rate in case of differential sampling during training
2. We introduce the 2D decision boundary optimization problem in terms of maximizing recall for a target precision (or vice versa) using both uncertainty and model score. Keeping in view computational efficiency, we partition the model score × uncertainty space into bins, demonstrate that the 2D decision boundary optimization problem is connected to bin-packing, and prove that it is NP-hard (for variable bin sizes) via reduction from the subset-sum problem (Caprara et al., 2000). This formulation is independent of the choice of modeling and uncertainty estimation method.
3. We present multiple algorithms for solving the 2D binned decision boundary problem defined over score and uncertainty derived from any blackbox classification model. We propose an equi-weight bin construction by considering uncertainty quantiles that are further split into score quantiles. For this case, we present a polynomial time DP algorithm with optimality guarantees. Additionally, we propose a greedy algorithm that first performs isotonic regression (Stout, 2013) independently for each uncertainty level, and selects a global threshold on calibrated probabilities.
4. We present results on three real-world datasets to demonstrate that our proposed 2D decision boundary algorithms yield 25%-40% gain in recall@precision over vanilla score-thresholding.

## 2   RELATED WORK

**Uncertainty Modeling.** Existing approaches for estimating uncertainty can be broadly categorized as Bayesian methods (Xu & Akella, 2008; Blundell et al., 2015b; Kendall & Gal, 2017), Monte

Carlo methods (Gal & Ghahramani, 2016) and ensembles (Lakshminarayanan et al., 2017). Dropout and ensemble methods estimate uncertainty by sampling probability predictions from different sub-models during inference, and are compute intensive. Recently, Charpentier et al. (2020) proposed Posterior Network to directly learn the posterior distribution over predicted probabilities, enabling fast uncertainty estimation for input samples in a single forward pass and providing an analytical framework for estimating both aleatoric and epistemic uncertainty, which makes it amenable for our analysis of score estimation bias. While Bengs et al. (2022) provides a detailed discussion on Posterior Networks highlighting the gaps in using learning with uncertain cross-entropy loss function to accurately estimate epistemic uncertainty, this aspect is orthogonal to our results since we only employ Posterior Network as an exemplar of uncertainty estimation.

**Uncertainty-based Decision Making.** (Blundell et al., 2015a) use uncertainty along with model score to drive explore-exploit style online-learning, but leveraging uncertainty to improve precision and recall has not been rigorously explored in the literature. Approaches in the digital pathology domain either use heuristics to define a 2D decision boundary using model score and estimated uncertainty (Pocevvciūtė et al., 2022), or use static uncertainty thresholds to withhold predictions for low-confidence samples from the test dataset, to boost model accuracy (Dolezal et al., 2022; Zhou et al., 2022). Troffaes (2007) and Denœux (2019) focus on decision-making using a risk-adjusted utility function that incorporates uncertainty modeled via belief functions and is determined by axiomatic criteria such as minmax and Hurwicz criterion without any data-based calibration.

**Model Score Recalibration.** These methods transform the model score into a well-calibrated probability using empirical observations on a hold-out set. Earlier approaches include histogram binning (Zadrozny & Elkan, 2001), isotonic regression (Stout, 2013), and temperature scaling (Guo et al., 2017), all of which consider the model score alone during recalibration. Uncertainty Toolbox (Chung et al., 2021) implements recalibration methods taking into account both uncertainty and model score but is currently limited to regression. In our work, we propose an algorithm (MIST 3) that first performs 1D-isotonic regression on samples within an uncertainty level to calibrate probabilities and then selects a global threshold. In addition to achieving a superior decision boundary, this results in lower calibration error compared to using score alone.

## 3 RELATIONSHIP BETWEEN SCORE ESTIMATION BIAS AND UNCERTAINTY

To demonstrate the dependence of score estimation bias on uncertainty, we consider a representative data generation scenario and a common uncertainty modeling method Posterior Network [2]

**Notation.** Let $\mathbf{x}$ denote an input point and $y$ the corresponding target label that takes values from the set of class labels $\mathcal{C} = \{0, 1\}$ with $c$ denoting the index over the labels. See Appendix I. We use $\mathbf{P}(\cdot)$ to denote probability and $[i]_{lb}^{ub}$ to denote an index iterating over integers in $\{lb, \cdots, ub\}$.

**3.1 Background: Posterior Network** Posterior Network (Charpentier et al., 2020) estimates a closed-form posterior distribution over predicted class probabilities for any new input sample via density estimation as described in Appendix D. For binary classification, the posterior distribution at $\mathbf{x}$ is a Beta distribution with parameters estimated by combining the model prior with pseudo-counts generated based on the learned normalized densities and observed class counts. Denoting the model prior and observed counts for the class $c \in \mathcal{C}$ by $\beta_c^P$ and $N_c$, the posterior distribution of predicted class probabilities at $\mathbf{x}$ is given by $q(\mathbf{x}) = \text{Beta}(\alpha_1(\mathbf{x}), \alpha_0(\mathbf{x}))$ where $\alpha_c(\mathbf{x}) = \beta_c^P + \beta_c(\mathbf{x})$ and $\beta_c(\mathbf{x}) = N_c \mathbf{P}(\mathbf{z}(\mathbf{x})|c; \phi), \forall c \in \mathcal{C}$. Here, $\mathbf{z}(\mathbf{x})$ is the penultimate layer representation of $\mathbf{x}$ and $\phi$ denotes parameters of a normalizing flow. Model score $S^{model}(\mathbf{x})$ for positive class is given by

$$S^{model}(\mathbf{x}) = \frac{\beta_1^P + \beta_1(\mathbf{x})}{\sum_{c \in \mathcal{C}}[\beta_c^P + \beta_c(\mathbf{x})]} = \frac{\alpha_1(\mathbf{x})}{\alpha_1(\mathbf{x}) + \alpha_0(\mathbf{x})}. \tag{1}$$

Uncertainty $u(\mathbf{x})$ for $\mathbf{x}$ is given by differential entropy of distribution $H(q(\mathbf{x}))$ [3]. Since $q(\mathbf{x})$ is Beta distribution, for same score, (i.e., $\alpha_1(\mathbf{x})/\alpha_0(\mathbf{x})$), uncertainty is higher when $\sum_{c \in \mathcal{C}} \alpha_c(\mathbf{x})$ is lower.

**3.2 Analysis of Score Estimation Bias**: For an input point $\mathbf{x}$, let $S^{true}(\mathbf{x}), S^{train}(\mathbf{x}), S^{test}(\mathbf{x})$, and $S^{model}(\mathbf{x})$ denote the true positivity, empirical positivity in the train and test sets, and the model

---

[2]Theorem 3.1(a) connecting train, true, and test positivity is independent of the uncertainty modeling.

[3]$H(q(\mathbf{x})) = \log \mathcal{B}(\alpha_0, \alpha_1) - (\alpha_0 + \alpha_1 - 2)\psi(\alpha_0 + \alpha_1) - \sum_{c \in \mathcal{C}}(\alpha_c - 1)\psi(\alpha_c)$ where $\psi(\cdot)$ is the digamma function and $\mathcal{B}(\cdot, \cdot)$ is the Beta function.

score respectively. Assuming the train and test sets are drawn from the same underlying distribution with possible differential sampling across the classes, these variables are dependent on each other as shown in Fig. 2. We consider the following generation mechanism (Appendix E) where the true positivity rate is sampled from a global beta prior, i.e., $S^{true}(\mathbf{x}) \sim Beta(\beta_1^T, \beta_0^T)$. The labels $y(\mathbf{x})$ in the test set are generated from Bernoulli distribution centered at $S^{true}(\mathbf{x})$. In the case of training set, we assume that the negative class is sampled at rate $\frac{1}{\tau}$ compared to positive class. Note that $\tau > 1$ indicates undersampling, and $\tau < 1$ indicates oversampling of the negative class. We define $\gamma(\mathbf{x}) = \frac{\beta_1^P + \beta_0^P}{\beta_1(\mathbf{x}) + \beta_0(\mathbf{x})}$, i.e., the ratio of combined priors to the combined likelihood evidence.

Using Bayes' rule, one can estimate the expected true and test positivity rates conditioned on train positivity in terms of available evidence. For Posterior Networks, one could further use the relationship between the train positivity rate and the model score to obtain the expected score estimation bias. For ease of presentation, we show the result for $\tau = 1$ in Theorem 3.1 below with the general case (Theorem E.4) and proof details in Appendix E.

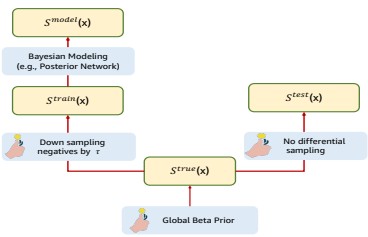

Figure 2: Dependencies among various positivity rates and the model score.

**Theorem 3.1.** *For data generated as per Fig. 2 but no differential sampling ($\tau = 1$), the below results hold:*
*(a) The expected test and true positivity rate conditioned on the train positivity are equal and given by*

$$E[S^{true}(\mathbf{x})|S^{train}(\mathbf{x})] = E[S^{test}(\mathbf{x})|S^{train}(\mathbf{x})] = \frac{S^{train}(\mathbf{x}) + \xi\lambda(\mathbf{x})}{1 + \lambda(\mathbf{x})}.$$

*where $\xi = \frac{\beta_1^T}{\beta_1^T + \beta_0^T}$ is the positive global prior, and $\lambda(\mathbf{x}) = \frac{\beta_1^T + \beta_0^T}{\beta_1(\mathbf{x}) + \beta_0(\mathbf{x})}$ is its ratio to evidence.*
*(b) For Posterior Networks, test and true positivity rate conditioned on model score $S^{model}(\mathbf{x})$ can be obtained using $S^{train}(\mathbf{x}) = S^{model}(\mathbf{x}) - (\omega - S^{model}(\mathbf{x}))\gamma(\mathbf{x})$. Hence, the estimation bias, i.e., difference between model score and test positivity is given by $\frac{(S^{model}(\mathbf{x})(\nu-1) + \omega - \xi\nu)\gamma(\mathbf{x})}{1 + \nu\gamma(\mathbf{x})}$, where $\omega = \frac{\beta_1^P}{\beta_1^P + \beta_0^P}$ and $\nu = \frac{\lambda(\mathbf{x})}{\gamma(\mathbf{x})} = \frac{\beta_1^T + \beta_0^T}{\beta_1^P + \beta_0^P}$ is the ratio of global and model priors.*

**Relationship of $\gamma(\mathbf{x})$ and $\mathbf{u}(x)$:** Note that $\sum_c \alpha_c(\mathbf{x}) = [\sum_c \beta_c^P](1 + \frac{1}{\gamma(\mathbf{x})})$. For a fixed score, $\sum_c \alpha_c(\mathbf{x})$ varies inversely with uncertainty $u(\mathbf{x}) = H(q(\mathbf{x}))$, making the latter positively correlated with $\gamma(\mathbf{x})$. Further details in Appendix E.2.

**No differential sampling ($\tau = 1$).** Since the model scores are estimated by combining the model priors and the evidence, $S^{model}(\mathbf{x}) = s(\mathbf{x})$ differs from the train positivity rate in the direction of the model prior ratio $\omega$. On the other hand, expected true and test positivity rate differ from train positivity rate in the direction of true class prior ratio $\xi$. When the model prior matches true class prior both on positive class ratio and magnitude, i.e., $\nu = 1, \xi = \omega$, there is no estimation bias. In practice, model priors are often chosen to have low magnitude and estimation bias is primarily influenced by global prior ratio with overestimation (i.e., expected test positivity < model score) in the higher score range

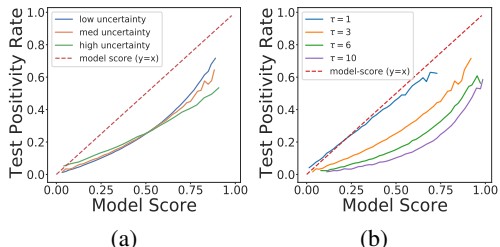

Figure 3: Test positivity vs. model score curves for (a) few choices of $\gamma(\mathbf{x})$ with $\omega = 0.5$, $\tau = 3$, and (b) few values of $\tau$ with $\omega = 0.5$ and medium uncertainty using data simulation as per Fig. 2.

($\xi < s(\mathbf{x})$) and the opposite is true when ($\xi > s(\mathbf{x})$). The extent of bias depends on relative strengths of priors w.r.t evidence denoted by $\gamma(\mathbf{x})$, which is correlated with uncertainty. For this case, the expected test positivity is linear and monotonically increasing in model score. The trend with respect to uncertainty depends on sign of $(s(\mathbf{x})(\nu - 1) + \omega - \xi\nu)$.

**General case ($\tau > 1$, Theorem E.4)** Here, the expected behavior is affected not only by the interplay of the model prior, true class prior and evidence as in case of $\tau = 1$, but also the differential sampling. While the first aspect is similar to the case $\tau = 1$, the second aspect results in overestimation across the entire score range with the extent of bias increasing with $\tau$. Fig. 3(a) shows

the expected positivity rate for a few different choices of $\gamma(\mathbf{x})$ and a fixed choice of $\omega = 0.5$ and $\tau = 10$ while Fig. 3(b) shows the variation with different choices of $\tau$. We validate this behavior by comparison with empirical observations in Sec. 6.

The primary takeaway from Theorems 3.1 and E.4 is that the score estimation bias depends on both score and uncertainty for common scenarios. For a given model score, different samples can correspond to different true positivity rates based on uncertainty level, opening an opportunity to improve the quality of the decision boundary by considering both score and uncertainty. However, a direct adjustment of model score based on Theorems 3.1 and E.4 is not feasible or effective since the actual prior and precise nature of distributional difference between test and train settings might not follow Fig. 2 or even be known (see Appendix H). Further, even when there is information on differential sampling rate used in training, class-conditional densities learned from sampled distributions tend to be different from original distribution especially over sparse regions.

## 4   2-D DECISION BOUNDARY PROBLEM

Given an input space $\mathcal{X}$ and binary labels $\mathcal{C}$, binary classification typically involves finding a mapping $\psi_{\mathbf{b}} : \mathcal{X} \to \mathcal{C}$ that optimizes application-specific performance. Fig. 4 depicts a typical supervised learning setting where labeled training data $\mathcal{D}_{train}$, created via differential sampling along classes, is used to learn a model. A hold-out labeled set $\mathcal{D}_{hold}$, disjoint from training and similar in distribution to deployment setting is used to construct a labeling function $\psi_{\mathbf{b}}$ based on model output.

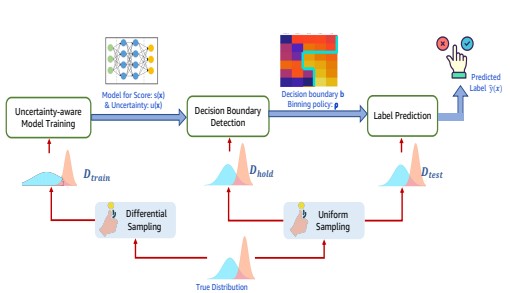

Figure 4: Binary classification with model training followed by decision boundary selection on hold-out set.

Traditionally, we use the labeling function $\psi_{\mathbf{b}}(\mathbf{x}) = \mathbb{1}[s(\mathbf{x}) \geq b]$ with boundary $\mathbf{b} = [b]$ defined in terms of a score threshold optimized based on the hold-out set $\mathcal{D}_{hold}$. When the model outputs both score and uncertainty $(s(\mathbf{x}), u(\mathbf{x}))$, we have a 2D space to be partitioned into positive and negative regions. In Sec. 3, we observed that the true positivity rate is monotonic with respect to score for a fixed uncertainty. Hence, we consider a boundary of the form $\psi_{\mathbf{b}}(\mathbf{x}) = \mathbb{1}[s(\mathbf{x}) \geq b(u(\mathbf{x}))]$, where $b(u)$ is the score threshold for uncertainty $u$.

To ensure tractability of decision boundary selection, a natural approach is to either limit $\mathbf{b}$ to a specific parametric family or discretize the uncertainty levels. We prefer the latter option as it allows generalization to multiple uncertainty estimation methods. Specifically, we partition the 2D score-uncertainty space into bins forming a grid such that the binning preserves the ordering over the space. (i.e., lower values go to lower level bins). This binning could be via independent splitting on both dimensions, or by partitioning on one dimension followed by a nested splitting on the other.

Let $\mathcal{S}$ and $\mathcal{U}$ denote the possible range of score and uncertainty values, respectively. Assuming $K$ and $L$ denote the desired number of uncertainty and score bins, let $\rho : \mathcal{U} \times \mathcal{S} \mapsto \{1, \cdots, K\} \times \{1, \cdots, L\}$ denote a partitioning such that any score-uncertainty pair $(u, s)$

$$\underset{\mathbf{b} \ \ \text{s.t.} \ \ precision(\psi_{\mathbf{b}}) \geq \sigma; \ 0 \leq b[i] \leq L}{\mathrm{argmax}} \quad recall(\psi_{\mathbf{b}}) \tag{2}$$

is mapped to a unique bin $(i, j) = (\rho^U(u), \rho^S(s))$ in the $K \times L$ grid. We capture relevant information from the hold-out set via two $K \times L$ matrices $[p(i, j)]$ and $[n(i, j)]$ where $p(i, j)$ and $n(i, j)$ denote the positive and the total number of samples in the hold-out set mapped to the bin $(i, j)$ in the grid. Using this grid representation, we now define the 2D Binned Decision Boundary problem. For concreteness, we focus on maximizing recall subject to a precision bound though our results can be generalized to other settings where the optimal operating point can be derived from the PR curve.

**2D Binned Decision Boundary Problem (2D-BDB):** Given a $K \times L$ grid of bins with positive sample counts $[p(i, j)]_{K \times L}$ and total sample counts $[n(i, j)]_{K \times L}$ corresponding to the hold-out set $D_{hold}$ and a desired precision bound $\sigma$, find the optimal boundary $\mathbf{b} = [b(i)]_{i=1}^{K}$ that maximizes recall subject to the precision bound as shown in Eqn. 2. Here $recall(\psi_{\mathbf{b}})$ and $precision(\psi_{\mathbf{b}})$ denote the recall and precision of the labeling function $\psi_b(\mathbf{x}) = \mathbb{1}[\rho^S(s(\mathbf{x})) > b(\rho^U(u(\mathbf{x})))]$ with

respect to true labels in $D_{hold}$. While $D_{hold}$ is used to determine the optimal boundary, actual efficacy is determined by performance on unseen test data.

**Connection to Knapsack Problem.** Note that the 2D decision boundary problem has similarities to the knapsack problem in the sense that given a set of items (i.e., bins), we are interested in choosing a subset that maximizes a certain "profit" aspect while adhering to a bound on a specific "cost" aspect. However, there are two key differences - 1) the knapsack problem has notions of cost and profit, while in our case we have precision and recall. On the other hand, our cost aspect is the false discovery rate (i.e., 1- precision) which is not additive, and the change in precision due to selection of a bin depends on previously selected bins, and 2) our problem setting has more structure since bins are arranged in a 2D-space with constraints on how these are selected.

## 5 2-D DECISION BOUNDARY ALGORITHMS

We provide results of computational complexity of 2D-BDB problem along with various solutions.

**5.1 NP-hardness Result**: It turns out the problem of computing optimal decision boundary over a 2D grid of bins (2D-BDB) is intractable for general case where bins have different sizes. We use a reduction from NP-hard subset-sum problem (Garey & Johnson, 1990) for the proof (Appendix F).

**Theorem 5.1.** *The problem of computing an optimal 2D-binned decision boundary is NP-hard.* □

**5.2 Equi-weight Binning Case** A primary reason for the intractability of 2D-BDB problem is that one cannot ascertain the relative "goodness" (i.e., recall subject to precision bound) of a pair of bins based on their positivity rates alone. For instance, it is possible that a bin A with lower positivity rate might be preferable to a bin B with higher positivity rate since the choice would be based on the overall positivity accounting for the current selected bins. Specifically, for current selection, bin A, bin B with total samples and positives given by $(N, P)$, $(n_A, p_A)$ and $(n_B, p_B)$, we could have $p_A/n_A > p_B/n_B$ and $(P + p_B)/(N + n_B) > (P + p_A)/(N + n_A)$ for $n_A \neq n_B)$. To address this, we propose a binning policy that preserves the partial ordering along score and uncertainty yielding equal-sized bins. We design an optimal algorithm for this special case using the fact that a bin with higher positivity is preferable among two bins of the same size.

**Binning strategy**: To construct an equi-weight $K \times L$ grid, we first partition the samples in $D_{hold}$ into $K$ quantiles along the uncertainty dimension and then split each of these into $L$ quantiles along the score. The bin indexed by $(i, j)$ contains samples from $i^{th}$ global uncertainty quantile and the $j^{th}$ score quantile local to $i^{th}$ uncertainty quantile. This mapping preserves the partial ordering that for any score level, the uncertainty bin indices are monotonic with respect to its actual values. Note that while this yields equal-sized bins on $D_{hold}$, using same boundaries on the similarly distributed test set will only yield approximately equal bins.

---

**Algorithm 1** Optimal Equi-weight DP-based Multi-Thresholds [EW-DPMT]

---

**Input:** Equi-sized $K \times L$ grid with positive sample counts $[p(i, j)]_{K \times L}$, total count $N$, precision level $\sigma$

**Output:** maximum (unnormalized) recall $R^*$ and corresponding optimal boundary $\mathbf{b}^*$ for precision $\geq \sigma$
  *// Initialization*
  $R(i, m) = -\infty; \quad b(i, m, i') = -1;$
  $\forall [i]_1^K, \; [i']_1^K, \; [m]_0^{KL}$
  *// Pre-computation of cumulative sums of positives*
  $\pi(i, 0) = 0, \quad [i]_1^K$
  $\pi(i, j) = \sum_{j'=L-j+1}^{L} p(i, j'), \quad [i]_1^K, \; [j]_1^L$
  *// Base Case: First Uncertainty Level*
  $R(1, m) = \pi(1, m); b(1, m, 1) = L - m, \quad [m]_0^L$
  *// Decomposition: Higher Uncertainty Levels*
  **for** $i = 2$ to $K$ **do**
     **for** $m = 0$ to $iL$ **do**
        $j^* = \underset{0 \leq j \leq L}{\mathrm{argmax}}[\pi(i, j) + R(i - 1, m - j)]$
        $R(i, m) = \pi(i, j^*) + R(i - 1, m - j^*)$
        $b(i, m, :) = b(i - 1, m - j^*, :)$
        $b(i, m, i) = L - j^*$
     **end for**
  **end for**
  *// Maximum Recall for Precision*
  $m^* = \underset{0 \leq m \leq KL \; s.t. \frac{KL}{mN} R(K, m) \geq \sigma}{\mathrm{argmax}} [R(K, m)]$
  $R^* = R(K, m^*); \mathbf{b}^* = b(K, m^*, )$
  **return** $(R^*, \mathbf{b}^*)$

---

**Dynamic Programming (DP) Algorithm**: For equi-weight binning, we propose a DP algorithm (Algorithm 1) for the 2D-BDB problem that identifies a maximum recall decision boundary for a given precision bound by constructing possible boundaries over increasing uncertainty levels. For $1 \leq i \leq K, 0 \leq m \leq KL$, let $R(i, m)$ denote the maximum true positives for any boundary over

the sub-grid with uncertainty levels up to the $i^{th}$ uncertainty level such that the boundary has exactly $m$ bins in its positive region. Further, let $b(i, m, :)$ denote the optimal boundary that achieves this maximum with $b(i, m, i')$ denoting the boundary position for the uncertainty level $i'(\leq i)$. Since bins are equi-sized, for a fixed positive bin count, the set with most positives yields the highest precision and recall. For the base case $i = 1$, a feasible solution exists only for $0 \leq m \leq L$ and corresponds to picking exactly $m$ bins, i.e., score threshold index $b(1, m, 1) = L - m$. For $i > 1$, we can decompose the estimation of maximum recall as follows. Let $j$ be the number of positive region bins from the $i^{th}$ uncertainty level. Then the budget available for the lower $(i - 1)$ uncertainty levels is exactly $m - j$. Hence, we have, $R(i, m) = \max_{0 \leq j \leq L} [\pi(i, j) + R(i - 1, m - j)]$,

where $\pi(i, j) = \sum_{j'=L-j+1}^{L} p(i, j')$, i.e., the count of positives in the $j$ highest score bins. The optimal boundary $b(i, m, :)$ is obtained by setting $b(i, m, i) = L - j^*$ and the remaining thresholds to that of $b(i - 1, m - j^*, :)$ where $j^*$ is the optimal choice of $j$ in the above recursion. Performing this computation progressively for all uncertainty levels and positive bin budgets yields maximum recall over the entire grid for each choice of bin budget. This is equivalent to obtaining the entire PR curve and permits choosing the optimal solution for a given precision bound. From $R(K, m)$, we can choose the largest $m$ that meets the desired input precision bound to achieve optimal recall. The overall computation time complexity is $O(K^2 L^2)$. More details in Appendix G.

**5.3 Other Algorithms** Even though the 2D-BDB problem with variable sized bins is NP-hard, it permits an optimal pseudo-polynomial time DP solution similar to the one presented above. VARIABLE-WEIGHT DP BASED MULTI-THRESHOLDS (VW-DPMT)) (4) tracks best recall at sample level instead of bin-level as in EW-DPMT (1). We also consider two greedy algorithms with lower computational complexity than the DP solution that are applicable to both variable and equal-size bins. The first, GREEDY-MULTI-THRESHOLD (GMT), computes score thresholds that maximize recall at a precision bound independently for each uncertainty level. The second, MULTI-ISOTONIC-SINGLE THRESHOLD (MIST) is based on recalibrating scores within each uncertainty level independently using 1-D isotonic regression. We identify a global threshold on calibrated probabilities that maximizes recall over the entire grid so that the precision bound is satisfied. Since the recalibrated scores are monotonic with respect to model score, the global threshold maps to distinct score quantile indices for each uncertainty level. This has a time complexity of $O(KL \log(KL))$.

# 6 EMPIRICAL EVALUATION

## 6.1 EXPERIMENTAL SETUP

**Datasets**: For evaluation, we use three binary classification datasets: (i) `Criteo`: An online advertising dataset consisting of ∼45 MM ad impressions with click outcomes, each with 13 continuous and 26 categorical features. We use the split of $72\% : 18\% : 10\%$ for train-validation-test from the benchmark, (ii) `Avazu`: Another CTR prediction dataset comprising ∼40 MM samples each with 22 features describing user and ad attributes. We use the train-validation-test splits of $70\% : 10\% : 20\%$, from the benchmark, (iii) `E-Com`: A proprietary e-commerce dataset with ∼4 MM samples where the positive class indicates a rare user action. We create train-validation-test sets in the proportion $50\% : 12\% : 38\%$ from different time periods. In all cases, we train with varying degrees of undersampling of negative class with test set as in the original distribution.

**Training**: For `Criteo` and `Avazu`, we use the SAM architecture (Cheng & Xue, 2021) as the backbone with 1 fully-connected layer and 6 radial flow layers for class distribution estimation. For `E-Com`, we trained a FT-Transformer (Gorishniy et al., 2021) backbone with 8 radial flow layers.
**Binning strategies**: We consider two options: (i) `Equi-span` where the uncertainty and score ranges are divided into equal sized $K$ and $L$ intervals, respectively. Samples with uncertainty in the $i^{th}$ uncertainty interval, and score in the $j^{th}$ score interval are mapped to bin $(i, j)$. (ii) `Equi-weight` where we first partition along uncertainty and then score as described in Sec. 5.
**Algorithms**: We compare our proposed decision boundary selection methods against (i) the baseline of using only score, SINGLE THRESHOLD (ST) disregarding uncertainty, and (ii) a state-of-the-art 2D decision boundary detection method for medical diagnosis (Pocevičiūtė et al., 2022), which we call HEURISTIC RECALIBRATION (HR). The greedy algorithms (GMT, MIST), variable weight DP algorithm (VW-DPMT) are evaluated on both `Equi-weight` and `Equi-span` settings, and the equi-weight DP algorithm (EW-DPMT) only on the former. All results are on the test sets.

| | Criteo, 90% Precision | | Avazu, 70% Precision | | E-Com, 70% Precision | |
|---|---|---|---|---|---|---|
| | $\tau=3$, Pos:Neg = 1:3 | | $\tau=5$, Pos:Neg = 1:5 | | $\tau=5$, Pos:Neg = 1:24 | |
| **Algorithm** | **Equi-Span** | **Equi-weight** | **Equi-Span** | **Equi-weight** | **Equi-Span** | **Equi-weight** |
| | *Score only* | | | | | |
| ST | $2.3\%_{\pm0.5\%}$ | $2.2\%_{\pm0.2\%}$ | $1.92\%_{\pm0.6\%}$ | $1.92\%_{\pm0.6\%}$ | $17.6\%_{\pm9.7\%}$ | $17.6\%_{\pm9.7\%}$ |
| | *Score and Uncertainty based* | | | | | |
| HR | $1.2\%_{\pm1.1\%}$ | $0.8\%_{\pm0.7\%}$ | $0.4\%_{\pm0.4\%}$ | $0.4\%_{\pm0.4\%}$ | $11.5\%_{\pm9.8\%}$ | $11.5\%_{\pm9.8\%}$ |
| GMT | $2.4\%_{\pm0.5\%}$ | $2.6\%_{\pm0.3\%}$ | $2.6\%_{\pm0.3\%}$ | $2.6\%_{\pm0.3\%}$ | $17.8\%_{\pm8.7\%}$ | $20.3\%_{\pm6.7\%}$ |
| MIST | $2.5\%_{\pm0.2\%}$ | $2.7\%_{\pm0.3\%}$ | $2.7\%_{\pm0.3\%}$ | $2.7\%_{\pm0.3\%}$ | $18.7\%_{\pm9.2\%}$ | $21.6\%_{\pm6.7\%}$ |
| EW-DPMT | - | $2.7\%_{\pm0.3\%}$ | - | $2.7\%_{\pm0.3\%}$ | - | $22.3\%_{\pm6.7\%}$ |
| VW-DPMT | $2.7\%_{\pm0.3\%}$ | $2.7\%_{\pm0.3\%}$ | $2.4\%_{\pm0.3\%}$ | $2.4\%_{\pm0.3\%}$ | $20.0\%_{\pm8.7\%}$ | $22.3\%_{\pm6.3\%}$ |

Table 1: Recall@PrecisionBound of various decision boundary methods on Criteo, Avazu & E-Com data.

## 6.2 RESULTS AND DISCUSSION

**RQ1: Estimation Bias Dependence on Score & Uncertainty.** From Sec. 3, we observe that the estimation bias and thus the test positivity rate is dependent on both uncertainty and the model score. Fig. 1 and Fig. 3 show the empirically observed behavior on the Criteo dataset and synthetic data generated as per Fig. 2 respectively with $\omega = 0.5$, $\tau = 3, \xi = 0.25$ in both cases. The observed empirical trends are broadly aligned with the theoretical expectations in Sec. 3 even though the assumption of a global Beta prior might not be perfectly valid. In particular, the separation between uncertainty levels is more prominent for the higher score range in these imbalanced datasets, pointing to the criticality of considering uncertainty for applications where high precision is desirable. To validate this further, we examine subsets of data where the algorithms EW-DPMT and ST differ on the decision boundary for 90% precision (with #score-bins = 500, #uncertainty-bins = 3) on Criteo dataset. We observe that the bin $[(s(\mathbf{x}), u(\mathbf{x})) = (0.984, 0)]$ with positivity rate 0.91 is labeled as positive by EW-DPMT but negative by ST, while the reverse is true for the bin $[(s(\mathbf{x}), u(\mathbf{x})) = (0.996, 0.667)]$ with a positivity rate 0.87. Note that $((s(\mathbf{x}), u(\mathbf{x}))$ are percentiles here. This variation of positivity with uncertainty for the same score substantiates the benefits of flexible 2D decision boundary estimation. More analysis of these bins in Appendix C.1.

**RQ2: Relative Efficacy of Decision boundary Algorithms.** Table 1 shows the recall at high precision bounds for various decision boundary algorithms on three large-scale datasets with 500 score and 3 uncertainty bins, averaged over 5 runs with different seeds. Since Avazu and E-com did not have feasible operating points at 90% precision, we measured recall@70% precision. Across all the datasets, we observe a significant benefit when incorporating uncertainty in the decision boundary selection (paired t-test significance p-values in Table 3). At 90% precision, EW-DPMT on Criteo is able to achieve a 22% higher recall (2.7% vs 2.2%) over ST. Similar behavior is observed on Avazu and E-com datasets where the relative recall lift is 42% and 26% respectively. Further, the Equi-weight binning results in more generalizable boundaries with the best performance coming from the DP algorithms (EW-DPMT, VW-DPMT) and the isotonic regression-based MIST. The heuristic baseline HR (Pocevivciūtė et al., 2022) performs poorly since it implicitly assumes that positivity rate monotonically increases with uncertainty for a fixed score. While both EW-DPMT and MIST took similar time ($\sim 100s$) for 500 score bins and 3 uncertainty bins, the run-time of the former increases significantly with increase in the bin count. Considering the excessive computation required for VW-DPMT, isotonic regression-based algorithm MIST and EW-DPMT seem to be efficient practical solutions. Results on statistical statistical significance and runtime comparison are in Appendix C.3 and Appendix C.4. Fig. 6(a) and Table 2 show the gain in recall for uncertainty-based 2D-decision boundary algorithms relative to the baseline algorithm ST highlighting that the increase is larger for high precision range and decreases as the precision level is reduced. Experiments with other uncertainty methods such as MC-Dropout (Gal & Ghahramani, 2016) (see Table 5 ) also point to some but not consistent potential benefits possibly because Posterior networks capture both epistemic and aleatoric uncertainty while MC-Dropout is restricted to the former.

**RQ3: Dependence on choice of bins and undersampling ratio.**: **Binning configuration.** Fig. 5(a) and 5(b) show how performance (Recall@PrecisionBound) of EW-DPMT varies with the number of uncertainty and score bins for Criteo and Avazu datasets. We observe a dataset dependent sweet-spot (marked by star) for the choice of bins. Too many bins can lead to overfitting of the decision boundary on the hold-out set that does not generalize well to test setting, while under-binning leads to low recall improvements on both hold-out and test sets.

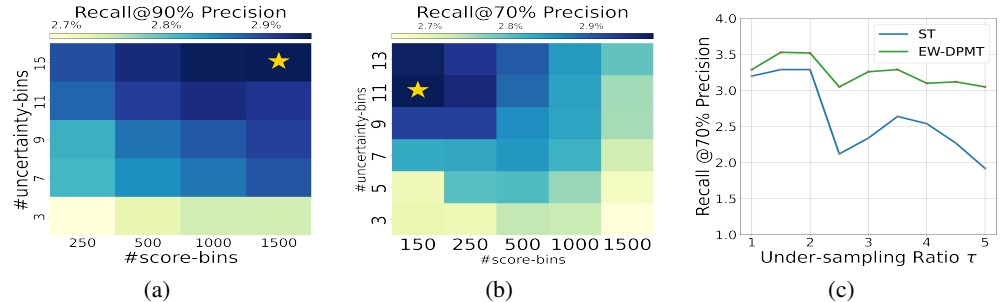

(a)                                    (b)                                    (c)

Figure 5: Impact of # bins along uncertainty and score for EW-DPMT on (a) `Criteo` ($\tau = 3$, Recall@90% Precision) and (b) `Avazu` ($\tau = 5$, Recall@70%Precision). (c) Impact of undersampling level ($\tau$) during training on Recall@70%Precision for ST and EW-DPMT on `Avazu`.

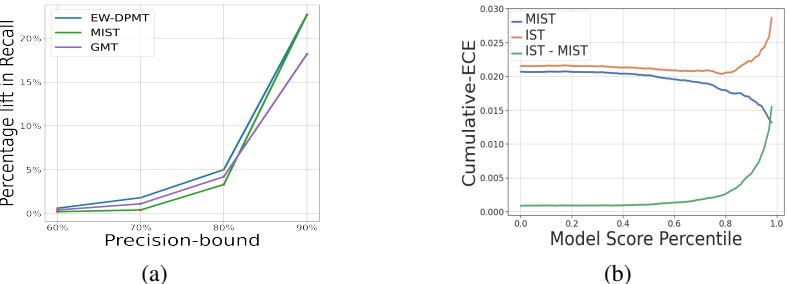

(a)                                                        (b)

Figure 6: (a) Relative gain in recall on uncertainty-based 2-D decision boundary algorithms over the baseline ST on `Criteo` dataset ($\tau = 3$) for different precision bounds. (b) Cumulative Expected Calibration Error(ECE) between MIST and IST baseline on `Avazu` dataset ($\tau = 5$).

**Undersampling Ratio.** Fig. 5(c) captures the Recall@70% Precision performance of EW-DPMT and ST for different levels of undersampling ($\tau$) of the negative class on the `Avazu` dataset averaged over 5 seeds. For both the algorithms, we observe an improvement in recall performance initially (till $\tau = 2.5$) which disappears for higher levels of downsampling in accordance with prior studies (Arjovsky et al., 2022). We observe that EW-DPMT consistently improves the Recall@70% precision over ST with more pronounced downsampling (i.e., higher values of $\tau$).

**RQ4: Impact of leveraging uncertainty for probability calibration.** To investigate the potential benefits of incorporating uncertainty in improving probability calibration, we compared the probabilities output from MIST algorithm with those from a vanilla isotonic regression (IST) baseline on Expected Calibration Error (ECE) for every score-bin, averaged across different uncertainty levels. Fig. 8 (b) demonstrates that the difference between ECE for MIST and IST increases as we move towards higher score range. Thus, the benefit of leveraging uncertainty estimates in calibration is more pronounced in high score range (i.e. at high precision levels). More details in Appendix C.6.

## 7    CONCLUSION

Leveraging uncertainty estimates for ML-driven decision-making is a key research area. In this paper, we examined potential benefits of utilizing uncertainty along with model score for binary classification. We provided theoretical analysis that points to the discriminating ability of uncertainty and formulated a novel 2-D decision boundary estimation problem based on score and uncertainty that is NP-hard. We also proposed practical solutions based on dynamic programming and isotonic regression. Empirical evaluation on real-world datasets point to the efficacy of utilizing uncertainty in improving classification performance. Future research directions include (a) designing efficient algorithms for joint optimization of binning configuration and boundary detection, (b) utilizing uncertainty for improving ranking performance and explore-exploit strategies in applications such as recommendations where the relative ranking matters and addressing data bias is critical, and (c) extensions to regression and multi-class classification settings (see Appendix H).

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

# A    REPRODUCIBILITY STATEMENT

To ensure the reproducibility of our experiments, we provide details of hyperparameters used for training posterior network model with details of model (backbone used and flow parameters) in Sec. 6.1. All models were trained on NVIDIA 16GB V100 GPU. We provide the pseudo code of binning and all algorithms implemented in Sec. 5 and Appendix G with details of bin-configuration in Sec 6.2. All binning and decision boundary related operations were performed on 4-core machine using Intel Xeon processor 2.3 GHz (Broadwell E5-2686 v4) running Linux. Moreover, we will publicly open-source our code later after we cleanup our code package and add proper documentation for it.

# B    ETHICS STATEMENT

Our work is in accordance with the recommended ethical guidelines. Our experiments are performed on three datasets, two of which are well-known click prediction Datasets (`Criteo`, `Avazu`) datasets in public domain. The third one is a proprietary dataset related to customer actions but collected with explicit consent of the customers while adhering to strict customer data confidentiality and security guidelines. The data we use is anonmyized by one-way hashing. Our proposed methods are targeted towards classification performance for any generic classifier and carry the risks common to all AI techniques.

# C    ADDITIONAL EXPERIMENTAL RESULTS

## C.1    BENEFITS OF 2D-DECISION BOUNDARY ESTIMATION

To anecdotally validate the benefits of 2D decision boundary estimation, we run the algorithms EW-DPMT and ST on `Criteo` dataset and examine bins where the algorithms differ on the decision boundary for 90% precision. As mentioned earlier, the bin (bin $A$) with $[(s(\mathbf{x}), u(\mathbf{x})) = (0.984, 0)]$ and positivity rate $0.91$ is included in the positive region by EW-DPMT but excluded by ST while the reverse is true for the bin (bin $B$) with $[(s(\mathbf{x}), u(\mathbf{x})) = (0.996, 0.667)]$ and positivity rate $0.87$. Note that $((s(\mathbf{x}), u(\mathbf{x}))$ are the score and uncertainty percentiles and not the actual values. We further characterise these bins using informative categorical features. Fig. 7 depicts pie-charts of the feature distribution of one of these features "C19" for both these bins as well as the corresponding score bins across all uncertainty levels and the entire positive region as identified by EW-DPMT. From the plots, we observe that the distribution of C19 for the positive region of EW-DPMT (Fig. 7 (a)) is similar to that of the bin A (Fig. 7 (b)) which is labeled positive by EW-DPMT and negative by ST and different from that of bin B (Fig. 7 (c)) that is labeled negative by EW-DPMT but positive by ST in terms of feature value V1 being more prevalent in the latter. We also observe that bins A and B diverge from the corresponding entire score bins across uncertainty-levels, i.e., Fig. 7 (c) and Fig. 7 (e) respectively. This variation of both feature distribution and positivity with uncertainty for the same score range highlights the need for flexible 2D decision boundary estimation beyond vanilla thresholding based on score alone.

## C.2    RECALL IMPROVEMENT AT DIFFERENT PRECISION LEVELS

To identify the precision regime where the 2D-decision boundary algorithms are beneficial, we measure the recall from the various algorithms for different precision bounds. Table 2 shows the results on CRITEO dataset ($\tau = 3$) highlighting that the relative improvement by leveraging uncertainty estimation in decision boundary estimation increases with precision bound. This empirically ties to the observation that the separation between different uncertainty levels is more prominent for higher score range, as this separation is used by 2D-decision boundary algorithms for improving recall.

## C.3    STATISTICAL SIGNIFICANCE OF VARIOUS ALGORITHMS VS ST

Table 3 captures the significance levels in the form of p-values on paired t-test (one-sided) comparing the different algorithms against the single-threshold (ST). It is evident that algorithms that leverage both score and uncertainty such as EW-DPMT, MIST, VW-DPMT and GMT significantly outperform ST, improving recall at fixed precision for all datasets.

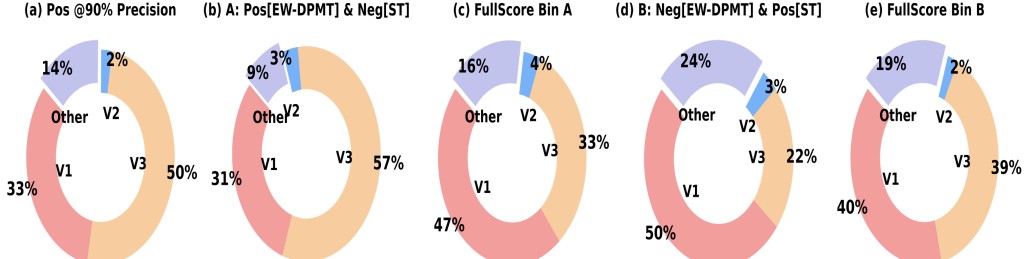

Figure 7: Distribution of subsets of data from `Criteo` with $\tau = 3$ across a key categorical feature (C19): (a) All positive samples as per the 90% precision decision boundary by EW-DPMT, (b) Bin A included by EW-DPMT in the positive region but excluded by ST, (c) Score bin corresponding to the bin A across all uncertainty levels, (d) Bin B excluded by EW-DPMT in the positive region but included by ST, and (e) Score bin corresponding to the bin B across all uncertainty levels. Here, V1 refers to C19 with value 1533924, V2 with 1533929, V3 with 1533925.

| Algorithm | Recall@ 60%Precision | Recall@ 70%Precision | Recall@ 80%Precision | Recall@ 90%Precision |
|---|---|---|---|---|
| *Score only* | | | | |
| ST | $46.6\%_{\pm 0.9\%}$ | $27.7\%_{\pm 1.1\%}$ | $12.0\%_{\pm 0.8\%}$ | $2.2\%_{\pm 0.2\%}$ |
| *Score and Uncertainty based* | | | | |
| GMT | $46.8\%_{\pm 0.9\%}$ **(+0.4%)** | $28.0\%_{\pm 1.0\%}$ **(+1.0%)** | $12.5\%_{\pm 0.8\%}$ **(+3.5%)** | $2.6\%_{\pm 0.3\%}$ **(+18.2%)** |
| MIST | $46.7\%_{\pm 0.9\%}$ **(+0.1%)** | $27.8\%_{\pm 1.0\%}$ **(+0.4%)** | $12.4\%_{\pm 0.6\%}$ **(+2.7%)** | $2.7\%_{\pm 0.3\%}$ **(+22.7%)** |
| EW-DPMT | $46.9\%_{\pm 0.8\%}$ **(+0.6%)** | $28.2\%_{\pm 1.0\%}$ **(+1.6%)** | $12.6\%_{\pm 0.8\%}$ **(+4.8%)** | $2.7\%_{\pm 0.3\%}$ **(+22.7%)** |

Table 2: Recall@ different precision levels for `Criteo` dataset ($\tau = 3$) for various decision boundary algorithms along with relative gains for each uncertainty level (in brackets) relative to the ST algorithm.

| | Criteo 90% Precision | | Avazu 70% Precision | | E-Com 70% Precision | |
|---|---|---|---|---|---|---|
| | $\tau$=3, Pos:Neg = 1:3 | | $\tau$=5, Pos:Neg = 1:5 | | $\tau$=5, Pos:Neg = 1:24 | |
| Algorithm | Equi-Span | Equi-weight | Equi-Span | Equi-weight | Equi-Span | Equi-weight |
| *Score and Uncertainty based* | | | | | | |
| ST vs. HR | 0.99 | 0.98 | 0.99 | 0.99 | | |
| ST vs. GMT | | 0.08 | 0.03 | 0.03 | 0.42 | 0.07 |
| ST vs. MIST | 0.05 | 0.03 | 0.02 | 0.02 | 0.1 | 0.03 |
| ST vs. EW-DPMT | - | 0.03 | - | 0.02 | - | 0.01 |
| ST vs. VW-DPMT | 0.03 | 0.03 | 0.02 | 0.02 | 0.02 | 0.01 |

Table 3: Significance level (p-values) of Paired t-test on Recall@PrecisionBound of different decision boundary algos on `Criteo`, `Avazu` and `E-Com` test datasets with #bins chosen same as per Table 1.

## C.4 RUNTIME OF VARIOUS ALGORITHMS

Table 4 shows the run-times (in seconds) for the best performing algorithms: MIST (Multi Isotonic regression Single score Threshold) and EW-DPMT (Equi-Weight Dynamic Programming based Multi threshold on `Criteo` ($\tau = 3$) dataset for different bin sizes with 64-core machine using Intel Xeon processor 2.3 GHz (Broadwell E5-2686 v4) running Linux. The runtimes are averaged over 5 experiment-seeds for each setting. The run-times do not include the binning time since this is the same for all the algorithms for a given binning configuration. It only includes the time taken to fit the decision boundary algorithm and obtain Recall@PrecisionBound.

Table 4: Wallclock runtime (in seconds) of various algorithms on `Criteo` Dataset ($\tau = 3$).

| Score-bins | 100 | | 500 | | 1000 | |
|---|---|---|---|---|---|---|
| Uncertainty-bins | EW-DPMT | MIST | EW-DPMT | MIST | EW-DPMT | MIST |
| 3 | 8 | 92 | 98 | 99 | 530 | 80 |
| 7 | 42 | 154 | 830 | 146 | 3434 | 99 |
| 11 | 106 | 233 | 1976 | 162 | 11335 | 99 |

From the theoretical analysis, we expect the runtime of decision boundary estimation for MIST to be $O(KL\log(KL))$. However, in practice there is a strong dependence only on $K$ i.e., the number of uncertainty bins since we invoke an optimized implementation of isotonic regression $K$ times. Furthermore, we perform the isotonic regression over the samples directly instead of the aggregates over the $L$ score bins which reduces the dependence on $L$. The final sorting that contributes to the $KL\log(KL)$ term is also optimized and does not dominate the run-time. For EW-DPMT, we expect a runtime complexity of $O(K^2L^2)$, i.e., quadratic in the number of bins. From decision-boundary algorithm fitting perspective, the observed run-times show faster than linear yet sub-quadratic growth due to fixed costs and python optimizations. Overall, MIST performs at par with EW-DPMT on the decision quality but takes considerably less time.

## C.5 RESULTS USING MC-DROPOUT

To understand the impact of choice of uncertainty estimation method, we report experiments on MC-Dropout (Gal & Ghahramani, 2016) algorithm in Table 1. MC-Dropout estimates epistemic uncertainty of a model by evaluating the variance in output from multiple forward passes of the model for every input sample. Resuts in Table 5 are from models trained for each dataset without any normalizing flow. While we observe substantial relative improvement when the recall is already low as in the case of `Avazu`, the magnitude of improvement is much smaller than in the case of Posterior Network possibly because the MC-Dropout uncertainty estimation does not account for aleatoric uncertainty.

| | `Criteo`, 90% Precision | `Avazu`, 70% Precision | `E-Com`, 70% Precision |
|---|---|---|---|
| | $\tau=3$, Pos:Neg = 1:3 | $\tau=5$, Pos:Neg = 1:5 | $\tau=5$, Pos:Neg = 1:24 |
| **Algorithm** | **Equi-weight** | **Equi-weight** | **Equi-weight** |
| *Score only* | | | |
| ST | $4.2\%_{\pm 0.1\%}$ | $1.28\%_{\pm 0.1\%}$ | $22.8\%_{\pm 4.7\%}$ |
| *Score and Uncertainty based* | | | |
| MIST | $4.3\%_{\pm 0.2\%}(\textbf{+2.4\%})$ | $1.8\%_{\pm 0.2\%}(\textbf{+40\%})$ | $23.1\%_{\pm 3.6\%}(\textbf{+1.3\%})$ |
| EW-DPMT | $4.3\%_{\pm 0.1\%}(\textbf{+2.4\%})$ | $1.9\%_{\pm 0.2\%}(\textbf{+48\%})$ | $23.7\%_{\pm 3.8\%}(\textbf{+3.9\%})$ |

Table 5: Performance of different decision boundary algorithms as measured Recall@PrecisionBound on `Criteo`, `Avazu` and `E-Com` test datasets with MC-Dropout as uncertainty estimation method.

## C.6 IMPACT OF USING UNCERTAINTY ESTIMATION ON CALIBRATION ERROR

For applications such as advertising, it is desirable to have well-calibrated probabilities and not just a decision boundary. To investigate the potential benefits of incorporating uncertainty in improving probability calibration, we compared the calibrated scores from MIST algorithm with those from a vanilla isotonic regression (IST) baseline. MIST fits a separate isotonic regression for each uncertainty level while IST fits a single vanilla isotonic regression on the model score. We evaluate the

Expected Calibration Error in the $j^{th}$ score-bin, $ECE@j$ as

$$ECE@j = \frac{1}{K} \sum_{i \in [1,K]} \frac{1}{n(i,j)} \left| \sum_{\mathbf{x} \in Bin(i,j)} (score[\mathbf{x}] - label[\mathbf{x}]) \right|,$$

where for each bin $(i,j)$, calibration error (CE) is evaluated on samples from the bin. CE is the absolute value of the average difference between the score and label for each sample $\mathbf{x} \in Bin(i,j)$, where $j \in [1, L], i \in [1, K]$. $n(i,j)$ is the number of samples in the $Bin(i,j)$. For both MIST and IST, we use the respective isotonic score for CE calculation. In Fig. 8 (a) and (c), we plot $ECE@j$ for all score bins for MIST and IST decision boundary algorithms on Criteo and Avazu datasets respectively, averaged over 5 different experiment seeds. The difference between $ECE@j$ MIST vs IST is pronounced at high-score levels, aligning with our primary observation that leveraging uncertainty estimates in decision boundary estimation helps improve recall at high precision levels.

We also define the cumulative-$ECE@j$ as the averaged calibration error for all bins with model-score percentile greater than $j$. The cumulative-$ECE@j$ results in a smoothened plot and the difference between the cumulative-$ECE@j$ for IST with that of MIST increases with model-score.

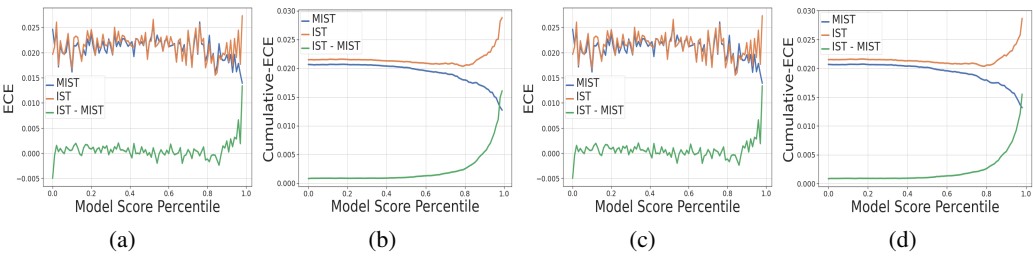

Figure 8: Impact of leveraging uncertainty in calibration by comparing MIST vs IST (a) $ECE@i$ for `Criteo` ($\tau = 3$) (b) Cumulative-$ECE@j$ for `Criteo` ($\tau = 3$) (c) $ECE@i$ for `Avazu` ($\tau = 5$) (d) Cumulative-$ECE@j$ for `Avazu` ($\tau = 5$).

### C.7 IMPACT OF UNCERTAINTY BASED MONOTONIC CONSTRAINTS

In the Equi-Weight DP (EW-DPMT) formulation, there is a separate threshold $b_i$ for each uncertainty level $i$. In other words, we assume monotonicity of the true conditional probability of the positive class with respect to the model score for a fixed uncertainty but not with respect to uncertainty for a fixed model score. This is because monotonicity with respect to uncertainty does not hold true across the entire score range. Fig. 3(a) shows the behavior (flipped trends for different score ranges) for the case of Beta priors with uncertainty estimated using Posterior Networks.

To empirically validate this observation, we consider EW-DPMT-MONO, a variant of the EW-DPMT algorithm where the decision boundary is constrained such that the true conditional probability is monotonically decreasing in uncertainty (i.e., $b_i \geq b_j$ for $i > j$ so that the boundary function looks like a staircase). This variant has higher computational complexity $O(K^2 L^3)$ than EW-DPMT where $K$ and $L$ are the number of uncertainty and score bins. Table 6 shows a comparison of the performance of the two algorithms for the same bin configuration. We observe that for high precision bound (as in `Criteo`), the monotoncity assumption does hold, but that is not true for lower precision bounds with the unconstrained version resulting in better performance. This aligns with the expected behavior in Fig. 3(a).

### C.8 IMPACT OF BINNING ALONG SCORE BEFORE UNCERTAINTY

In Sec. 5, we introduced the equi-weight nested binning strategy where samples in $D_{hold}$ are first partitioned into $K$ quantiles along the uncertainty dimension with each of these $K$ quantiles further split into $L$ quantiles along the score dimension. It is also possible to adopt an alternative binning strategy based on partitioning first on the score followed by uncertainty. Table 7 captures the Recall@Precision bound using the two equi-weight binning strategies for various 2-D decision boundary algorithms indicating that both the strategies yield similar results.

|  | Criteo, $\tau$=3, 90% Precision | Avazu, $\tau$=5, 70% Precision |
|---|---|---|
| *Score only* | | |
| ST | $2.2\%_{\pm 0.2\%}$ | $1.9\%_{\pm 0.6\%}$ |
| *Score and Uncertainty based* | | |
| EW-DPMT | $2.6\%_{\pm 0.3\%}$ | $2.6\%_{\pm 0.3\%}$ |
| EW-DPMT - MONO | $2.6\%_{\pm 0.6\%}$ | $1.9\%_{\pm 0.6\%}$ |

Table 6: Performance with different unconstrained and monotonic variants of EW-DPMT with uncertainty bins = 100 and score bins = 3. Note that here we used a smaller number of uncertainty bins instead of 500 (in the paper) to reduce computational effort.

| | Criteo, $\tau$=3, 90% Precision | | Avazu, $\tau$=5, 70% Precision | |
|---|---|---|---|---|
| Algorithm | Score-Unc (500,3) | Unc-Score (3,500) | Score-Unc (500,3 | Unc-Score (3,500) |
| *Score only* | | | | |
| ST | $2.2\%_{\pm 0.2\%}$ | $2.2\%_{\pm 0.2\%}$ | $1.9\%_{\pm 0.6\%}$ | $1.9\%_{\pm 0.6\%}$ |
| *Score and Uncertainty based* | | | | |
| MIST | $2.6\%_{\pm 0.2\%}$ | $2.6\%_{\pm 0.3\%}$ | $2.9\%_{\pm 0.2\%}$ | $2.6\%_{\pm 0.3\%}$ |
| GMT | $2.6\%_{\pm 0.4\%}$ | $2.7\%_{\pm 0.3\%}$ | $2.9\%_{\pm 0.2\%}$ | $2.7\%_{\pm 0.3\%}$ |
| EW-DPMT | $2.7\%_{\pm 0.2\%}$ | $2.7\%_{\pm 0.3\%}$ | $2.9\%_{\pm 0.2\%}$ | $2.7\%_{\pm 0.3\%}$ |

Table 7: Performance with different equi-weight binning strategies. Score-Unc involves splitting on score quantiles followed by that of uncertainty while it is the opposite for Unc-Score. Same number of score and uncertainty bins were used for both experiments.

## D  POSTERIOR NETWORKS

Posterior Network (PostNet) (Charpentier et al., 2020) builds on the idea of training a model to predict the parameters of the posterior distribution for each input sample. For classification, the posterior distribution (assuming conjugacy with exponential family distribution) would be Dirichlet distribution, and PostNet estimates the parameters of this distribution using Normalising Flows.

They model this by dividing the network into two components:

- **Encoder**: For every input $\mathbf{x}$, encoder ($f_\theta$) computes $z = f_\theta(\mathbf{x})$, a low-dimensional latent representation of the input sample in a high-dimensional space, capturing relevant features for classification. The encoder also yields sufficient statistics of the likelihood distribution in the form of affine-transform of $z(\mathbf{x})$ followed by application of log-softmax. Instead of learning a single-point classical softmax output, it learns a posterior distribution over them, characterized by Dirichlet distribution.
- **Normalizing flow (NF)**: This models normalized probability density $p(z|c, \phi)$ per class on the latent space $z$, intuitively acting as class conditionals in the latent space. The ground truth label counts along with normalized densities are used to compute the final pseudo counts. Thus, the component yields the likelihood evidence that is then combined with the prior to obtain the posterior for each sample.

The model is trained using an uncertainty aware formulation of cross-entropy. Here $\theta$ and $\phi$ are the parameters of the encoder and the NF respectively. Since both the encoder network $f_\theta$ and the normalizing flow parameterized by $\phi$ are fully differentiable, we can learn their parameters jointly in an end-to-end fashion. $q(\mathbf{x})$ is the estimated posterior distribution over $p(\mathbf{y}|\mathbf{x})$. The model's final classification prediction is the expected sufficient statistic and the uncertainty is the differential entropy of the posterior distribution. The model is optimised using stochastic gradient descent using loss function that combines cross entropy with respect to true labels and the entropy of $q(\mathbf{x})$.

## E  ESTIMATION BIAS ANALYSIS: PROOFS OF THEOREMS

### E.1  DATA GENERATION PROCESS

The true positivity rate $S^{true}(\mathbf{x})$ is generated from a global Beta prior with parameters $\beta_1^T$ and $\beta_0^T$, i.e.,

$$S^{true}(\mathbf{x}) \sim Beta(\beta_1^T, \beta_0^T).$$

The train and test samples at an input region $\mathbf{x}$ (modeled in a discrete fashion) are generated from the true positivity rate following a Bernoulli distribution with the negative train samples being undersampled by factor $\tau$. Let $N^{train}(\mathbf{x})$ and $N^{test}(\mathbf{x})$ denote the number of train and test samples at $\mathbf{x}$. Let $N_c^{train}(\mathbf{x})$ and $N_c^{test}(\mathbf{x})$, $c \in \{0, 1\}$ denote the class-wise counts. The positive counts for the train and test count are given by

$$N_1^{test}(\mathbf{x}) \sim Binomial(N^{test}(\mathbf{x}), S^{true}(\mathbf{x}))$$

$$N_1^{train}(\mathbf{x}) \sim Binomial(N^{train}(\mathbf{x}), \frac{\tau S^{true}(\mathbf{x})}{(\tau - 1)S^{true}(\mathbf{x}) + 1}).$$

The train and test positivity rates are given by $S^{train}(\mathbf{x}) = \frac{N_1^{train}(\mathbf{x})}{N^{train}(\mathbf{x})}$ and $S^{test}(\mathbf{x}) = \frac{N_1^{test}(\mathbf{x})}{N^{test}(\mathbf{x})}$. The model score $S^{model}(\mathbf{x})$ is obtained by fitting a model on the train set with no additional dependence on the test and true positivity rates. Fig. 2 shows the dependencies among the different variables.

**Lemma E.1.** *The relationship between train positivity $S^{train}(\mathbf{x})$ and model score for positive class $S^{model}(\mathbf{x})$ from Posterior Network is given by*

$$S^{train}(\mathbf{x}) = S^{model}(\mathbf{x}) - (\omega - S^{model}(\mathbf{x}))\gamma(\mathbf{x}).$$

*where*

- $\omega = \frac{\beta_1^P}{\beta_1^P + \beta_0^P}$

- $\gamma(\mathbf{x}) = \frac{\beta_1^P + \beta_0^P}{\beta_1(x) + \beta_0(x)}$

*Proof.* Using the notation in Sec. 3, the pseudo-counts $\beta_c(\mathbf{x})$, $c \in \{0, 1\}$ correspond to the observed positive and negative counts at $\mathbf{x}$. Hence, the train positivity is given by

$$S^{train}(\mathbf{x}) = \frac{\beta_1(\mathbf{x})}{\beta_1(\mathbf{x}) + \beta_0(\mathbf{x})}.$$

This gives us $\beta_0(\mathbf{x}) = \beta_1(\mathbf{x})\left(\frac{1 - S^{train}(\mathbf{x})}{S^{train}(\mathbf{x})}\right)$.

Using the definitions of $\omega$ and $\gamma(\mathbf{x})$, the model score $S^{model}(\mathbf{x})$ from Posterior Network (Eqn. 1 can now be expressed in terms of $\omega$, $S^{train}(\mathbf{x})$ and $\gamma(\mathbf{x})$ as follows:

$$S^{model}(\mathbf{x}) = \frac{\beta_1^P + \beta_1(\mathbf{x})}{\sum_{c \in \mathcal{C}}[\beta_c^P + \beta_c(\mathbf{x})]} = \frac{\omega\gamma(\mathbf{x}) + S^{train}(\mathbf{x})}{1 + \gamma(\mathbf{x})}.$$

Hence, $S^{train}(\mathbf{x}) = S^{model}(\mathbf{x}) - (\omega - S^{model}(\mathbf{x}))\gamma(\mathbf{x})$. $\qquad\square$

**Theorem E.2.** *For the case where data is generated as per Fig. 2 and negative class is undersampled at the rate $\frac{1}{\tau}$, the following results hold:*
*(a) The expected true positivity rate conditioned on the train positivity is given by the expectation of the distribution,*

$$Q(r) = \frac{C}{(1 + (\tau - 1)r)^n} Beta(n(\xi\lambda(\mathbf{x}) + S^{train}(\mathbf{x})), n((1 - \xi)\lambda(\mathbf{x}) + 1 - S^{train}(\mathbf{x}))).$$

- $n = \beta_1(\mathbf{x}) + \beta_0(\mathbf{x})$ *denotes evidence, $C$ is a normalizing constant, $\xi = \frac{\beta_1^T}{\beta_1^T + \beta_0^T}$ is the positive global prior, and $\lambda(\mathbf{x}) = \frac{\beta_1^T + \beta_0^T}{\beta_1(\mathbf{x}) + \beta_0(\mathbf{x})}$ is the ratio of global priors to evidence.*

*(b) When there is no differential sampling, i.e., $\tau = 1$, the expectation has a closed form and is given by*

$$E[S^{true}(\mathbf{x})|S^{train}(\mathbf{x})] = \frac{S^{train}(\mathbf{x}) + \xi\lambda(\mathbf{x})}{1 + \lambda(\mathbf{x})}.$$

*Proof.* Let $N^{train}(\mathbf{x})$ and $N_1^{train}(\mathbf{x})$ denote the number of train samples and positive samples associated with any input region $\mathbf{x}$. Then the train positivity $S^{train}(\mathbf{x}) = \frac{N_1^{train}(\mathbf{x})}{N^{train}(\mathbf{x})}$.

Since $N^{train}(\mathbf{x})$ corresponds to the probability mass and pseudo counts at $\mathbf{x}$, we consider regions with a fixed size $N^{train}(\mathbf{x}) = n$. The expected true positivity rate for all $\mathbf{x}$ with size $N^{train}(\mathbf{x}) = n$ conditioned on $S^{train}(\mathbf{x}) = \frac{k}{n}$ is given by $E[S^{true}(\mathbf{x})|S^{train}(\mathbf{x}) = k/n] = E[S^{true}(\mathbf{x})|N_1^{train}(\mathbf{x}) = k]$.

For brevity, we omit the explicit mention of the dependence on $\mathbf{x}$ for variables $S^{model}(\mathbf{x})$, $S^{train}(\mathbf{x})$, $S^{test}(\mathbf{x})$, and $S^{true}(\mathbf{x})$.

The conditional probability $p(S^{true}|N_1^{train} = k)$ is given by the Bayes rule. Specifically, we have

$$p(S^{true} = r|N_1^{train} = k) = \frac{p(S^{true} = r)p(N_1^{train} = k|S^{true} = r)}{p(N_1^{train} = k)}.$$

Here $S^{true}$ follows a global Beta prior and $N_1^{train}$ a Binomial distribution with downsampling of negative examples at the rate $\frac{1}{\tau}$. For $S_{true} = r$, the success probability of the Binomial distribution (probability of obtaining a sample with $y = 1$) is given by $\frac{r}{r + \frac{1-r}{\tau}} = \frac{\tau r}{(1 + (\tau - 1)r)}$. Hence,

$$
\begin{aligned}
p(S^{true} = r)p(N_1^{train} = k|S^{true} = r) &= Beta(\beta_1^T, \beta_0^T)\binom{n}{k}\left(\frac{\tau r}{1 + (\tau - 1)r}\right)^k \left(\frac{1-r}{1 + (\tau - 1)r}\right)^{n-k} \\
&= \frac{C_0 \tau^k}{(1 + (\tau - 1)r)^n} Beta(\beta_1^T + k, \beta_0^T + n - k),
\end{aligned}
$$

where $C_0$ is a normalizing constant independent of $r$ and $\tau$. While the integral $\int_r p(S^{true} = r)p(N_1^{train} = k|S^{true} = r)dr$ over $[0, 1]$ does not have a closed form, we do observe that the desired conditioned distribution will have a similar form with a different normalizing constant $C$ since the denominator is independent of $r$:

$$
\begin{aligned}
p(S^{true} = r|N_1^{train} = k) &= \frac{p(S^{true} = r)p(N_1^{train} = k|S^{true} = r)}{\int_r p(S^{true} = r)p(N_1^{train} = k|S^{true} = r)dr} \\
&= \frac{C}{(1 + (\tau - 1)r)^n} Beta(\beta_1^T + k, \beta_0^T + n - k).
\end{aligned}
$$

The expected true positivity rate conditioned on $N_1^{train} = k$ is the mean of this new distribution, which does not have a closed form but can be numerically computed and will be similar to the simulation results in Fig. 3(b).

Using the definitions of $\xi = \frac{\beta_1^T}{\beta_1^T + \beta_0^T}$ and $\lambda(\mathbf{x}) = \frac{\beta_1^T + \beta_0^T}{\beta_1(\mathbf{x}) + \beta_0(\mathbf{x})}$, we can rewrite $\beta_1^T = n\xi\lambda(\mathbf{x})$ and $\beta_0^T = n(1 - \xi)\lambda(\mathbf{x})$. Further, observing that $S^{train}(\mathbf{x}) = k/n$, we can express this distribution as

$$Q(r) = \frac{C}{(1 + (\tau - 1)r)^n} Beta(n(\xi\lambda(\mathbf{x}) + S^{train}(\mathbf{x})), n((1 - \xi)\lambda(\mathbf{x}) + 1 - S^{train}(\mathbf{x}))),$$

which yields the desired result.

*Part b:* For the case where $\tau = 1$, the term $\frac{1}{(1 + (\tau - 1)r)^n} = 1$ and the distribution $Q(r)$ reduces to just the Beta distribution $Beta(n(\xi\lambda(\mathbf{x}) + S^{train}(\mathbf{x})), n((1 - \xi)\lambda(\mathbf{x}) + 1 - S^{train}(\mathbf{x})))$. Note the normalizing constant $C = 1$ since the Beta distribution itself integrates to 1. The expected true positivity is the just the mean of this Beta distribution, i.e.,

$$
\begin{aligned}
E[S^{true}(\mathbf{x})|S^{train}(\mathbf{x})] &= \frac{n(\xi\lambda(\mathbf{x}) + S^{train}(\mathbf{x}))}{n(\xi\lambda(\mathbf{x}) + S^{train}(\mathbf{x})) + n((1 - \xi)\lambda(\mathbf{x}) + 1 - S^{train}(\mathbf{x}))} \\
&= \frac{S^{train}(\mathbf{x}) + \xi\lambda(\mathbf{x})}{1 + \lambda(\mathbf{x})}.
\end{aligned}
$$

$\square$

**Theorem E.3.** *For the case where data is generated as per Sec. E.1, the expected test and true positivity rate conditioned either on the train positivity rate or model score for positive class are equal, i.e.,*

$$E[S^{test}(\mathbf{x})|S^{train}(\mathbf{x})] = E[S^{true}(\mathbf{x})|S^{train}(\mathbf{x})],$$

$$E[S^{test}(\mathbf{x})|S^{model}(\mathbf{x})] = E[S^{true}(\mathbf{x})|S^{model}(\mathbf{x})].$$

*Proof.* From the data generation process in Sec. E.1, we observe that the test label samples $Y_{test}(\mathbf{x})$ at the input region $\mathbf{x}$ are generated by Bernoulli distribution centered around $S^{true}(\mathbf{x})$ i.e., $Y_{test}(\mathbf{x}) \sim Bernoulli(S^{true}(\mathbf{x}))$ and $S^{test}$ is the mean of $Y^{test}$ over the test samples.

Hence, the test labels $Y^{test}(\mathbf{x})$ and test positivity rate $S^{test}(\mathbf{x})$ are independent of the model score $S^{model}(\mathbf{x})$ and $S^{train}(\mathbf{x})$ given the $S^{true}(\mathbf{x})$. For brevity, we omit the explicit mention of the dependence on $\mathbf{x}$ for variables $Y^{test}(\mathbf{x})$, $S^{model}(\mathbf{x})$, $S^{train}(\mathbf{x})$, $S^{test}(\mathbf{x})$, and $S^{true}(\mathbf{x})$.

As $Y^{test}$ is conditionally independent of $S^{train}$ given $S^{true}$, we observe that

$$E[Y^{test}|S^{train}] = E_{S^{true}|S^{train}}[E[Y^{test}|S^{true}]].$$

However, since $Y_{test} \sim Bernoulli(S^{true}(\mathbf{x}))$, we have $E[Y^{test}|S^{true}] = S^{true}$. Therefore,

$$E[Y^{test}|S^{train}] = E_{S^{true}|S^{train}}[E[Y^{test}|S^{true}]]. = E[S^{true}|S^{train}]$$

Since $S^{test}$ is itself the expectation over $Y_{test}$, by the law of iterated expectations, we have,

$$E[S^{test}|S^{train}] = E[Y^{test}|S^{train}] = E[S^{true}|S^{train}],$$

which is the desired result. The same result holds true even when conditioning on the model score $S^{model}$ since $Y^{test}$ and $S^{test}$ are also conditionally independent of $S^{model}$ given $S^{true}$.

$\square$

**Theorem E.4.** *For the case where data is generated as per Fig. 2 and negative class is undersampled at the rate $\frac{1}{\tau}$:*
*(a) The expected test and true positivity rate conditioned on the train positivity are equal and correspond to the expectation of the distribution,*

$$Q(r) = \frac{C}{(1 + (\tau - 1)r)^n} Beta(n(\xi\lambda(\mathbf{x}) + S^{train}(\mathbf{x})), n((1 - \xi)\lambda(\mathbf{x}) + 1 - S^{train}(\mathbf{x}))).$$

*When there is no differential sampling, i.e., $\tau = 1$, the expectation has a closed form and is given by*

$$E[S^{true}(\mathbf{x})|S^{train}(\mathbf{x})] = E[S^{test}(\mathbf{x})|S^{train}(\mathbf{x})] = \frac{S^{train}(\mathbf{x}) + \xi\lambda(\mathbf{x})}{1 + \lambda(\mathbf{x})}.$$

- $n = \beta_1(\mathbf{x}) + \beta_0(\mathbf{x})$ *denotes evidence, $C$ is a normalizing constant, $\xi = \frac{\beta_1^T}{\beta_1^T + \beta_0^T}$ is the positive global prior, and $\lambda(\mathbf{x}) = \frac{\beta_1^T + \beta_0^T}{\beta_1(\mathbf{x}) + \beta_0(\mathbf{x})}$ is the ratio of global priors to evidence.*

*(b) For Posterior Networks, the test and true positivity rate conditioned on the model score $S^{model}(\mathbf{x})$ can be obtained using $S^{train}(\mathbf{x}) = S^{model}(\mathbf{x}) - (\omega - S^{model}(\mathbf{x}))\gamma(\mathbf{x})$. For $\tau = 1$, the estimation bias, i.e., difference between model score and test positivity is given by $\frac{(S^{model}(\mathbf{x})(\nu - 1) + \omega - \xi\nu)\gamma(\mathbf{x})}{1 + \nu\gamma(\mathbf{x})}$.*

- $\omega = \frac{\beta_1^P}{\beta_1^P + \beta_0^P}$ *and $\nu = \frac{\lambda(\mathbf{x})}{\gamma(\mathbf{x})} = \frac{\beta_1^T + \beta_0^T}{\beta_1^P + \beta_0^P}$ is the ratio of global and model priors.*

*Proof. Part a:*
From Theorem E.2, we directly obtain the result on the expectation of true positivity rate in terms of the train positivity both for the general case where $\tau \neq 1$ and for the special case of $\tau = 1$. Further, from Theorem E.3, we observe that the expected true positivity is also the same as the expected test positivity conditioned on the train positivity, which yields the desired result.

*Part b:*
From Lemma E.1, we obtain the relationship between the train positivity and the model score, i.e., $S^{train}(\mathbf{x}) = S^{model}(\mathbf{x}) - (\omega - S^{model}(\mathbf{x}))\gamma(\mathbf{x})$. which can be used to expression the expected train and test positivity directly in terms of the model score.

For the case $\tau = 1$, in particular, since $S^{train}(\mathbf{x})$ is deterministic function of $S^{model}(\mathbf{x})$ for a fixed $\gamma(\mathbf{x})$, we observe that

$$E[S^{test}(\mathbf{x})|S^{train}(\mathbf{x})] = \frac{S^{train}(\mathbf{x}) + \xi\lambda(\mathbf{x})}{1 + \lambda(\mathbf{x})}.$$

Expressing this in terms of $S^{model}(\mathbf{x})$ and $\gamma(\mathbf{x}) = \lambda(\mathbf{x})/\nu$ gives us

$$E[S^{test}(\mathbf{x})|S^{model}(\mathbf{x})] = \frac{S^{model}(\mathbf{x}) + (S^{model}(\mathbf{x}) + \xi\nu - \omega)\gamma(\mathbf{x})}{1 + \nu\gamma(\mathbf{x})}.$$

Thus, the estimation bias is given by

$$S^{model}(\mathbf{x}) - E[S^{test}(\mathbf{x})|S^{model}(\mathbf{x})] = \frac{(S^{model}(\mathbf{x})(\nu - 1) + \omega - \xi\nu)\gamma(\mathbf{x})}{1 + \nu\gamma(\mathbf{x})}.$$

$\square$

**Theorem E.5. [Restatement of Theorem 3.1]** *For data generated as per Fig. 2 but no differential sampling (i.e., $\tau = 1$), the following results hold:*
*(a) The expected test and true positivity rate conditioned on the train positivity are equal and given by*

$$E[S^{true}(\mathbf{x})|S^{train}(\mathbf{x})] = E[S^{test}(\mathbf{x})|S^{train}(\mathbf{x})] = \frac{S^{train}(\mathbf{x}) + \xi\lambda(\mathbf{x})}{1 + \lambda(\mathbf{x})}.$$

*where $\xi = \frac{\beta_1^T}{\beta_1^T + \beta_0^T}$ is the positive global prior, and $\lambda(\mathbf{x}) = \frac{\beta_1^T + \beta_0^T}{\beta_1(\mathbf{x}) + \beta_0(\mathbf{x})}$ is the ratio of global priors to evidence.*
*(b) For Posterior Networks, test and true positivity rate conditioned on model score $S^{model}(\mathbf{x})$ can be obtained using $S^{train}(\mathbf{x}) = S^{model}(\mathbf{x}) - (\omega - S^{model}(\mathbf{x}))\gamma(\mathbf{x})$. Hence, the estimation bias, i.e. difference between model score and test positivity is given by $\frac{(S^{model}(\mathbf{x})(\nu-1)+\omega-\xi\nu)\gamma(\mathbf{x})}{1+\nu\gamma(\mathbf{x})}$, where $\omega = \frac{\beta_1^P}{\beta_1^P + \beta_0^P}$ and $\nu = \frac{\lambda(\mathbf{x})}{\gamma(\mathbf{x})} = \frac{\beta_1^T + \beta_0^T}{\beta_1^P + \beta_0^P}$ is the ratio of global and model priors.*

*Proof.* Current claims are a special case of Theorem E.4 when $\tau = 1$. $\square$

### E.2 RELATIONSHIP OF $\gamma(\mathbf{x})$ AND $\mathbf{u}(x)$

From Theorem 3.1 of Sec. 3, we find that the model score estimation bias, (i.e., difference between model score and test positivity) for the case of no differential sampling ($\tau = 1$) is given by

$$\frac{(S^{model}(\mathbf{x})(\nu - 1) + \omega - \xi\nu)\gamma(\mathbf{x})}{1 + \nu\gamma(\mathbf{x})}.$$

Here, the bias depends on the model score, multiple constants $\xi = \frac{\beta_1^T}{\beta_1^T + \beta_0^T}$, $\omega = \frac{\beta_1^P}{\beta_1^P + \beta_0^P}$, $\nu = \frac{\lambda(\mathbf{x})}{\gamma(\mathbf{x})} = \frac{\beta_1^T + \beta_0^T}{\beta_1^P + \beta_0^P}$ and a variable quantity $\gamma(\mathbf{x}) = \frac{\beta_1^P + \beta_0^P}{\beta_1(\mathbf{x}) + \beta_0(\mathbf{x})}$, which is the ratio of model's prior evidence to the combined likelihood evidence for a sample $\mathbf{x}$.

As mentioned earlier in Sec. 3, $\gamma(\mathbf{x})$ is inversely correlated to the sum of the parameters $(\alpha_1(\mathbf{x}), \alpha_0(\mathbf{x}))$ of the epistemic Beta distribution $q(x)$ of the Posterior Network. Specifically, $\sum_c \alpha_c(\mathbf{x}) = [\sum_c \beta_c^P](1 + \frac{1}{\gamma(\mathbf{x})})$. Increasing $\gamma(x)$ corresponds to a decreasing $\sum_c \alpha_c(\mathbf{x})$, which in turn results in higher entropy $H(q(x))$ for the epistemic distribution, i.e., higher uncertainty $u(x)$. Fig. 9 depicts the monotonic relationship between uncertainty $u(\mathbf{x}) = H(q(x))^4$ and $\gamma(\mathbf{x})$ for various fixed values of model score and varying amount of evidence computed using a numerical simulation. Here, the global prior parameters are chosen as $(\beta_1^P, \beta_0^P) = (5, 10)$ and the total evidence $(\beta_0(\mathbf{x}) + \beta_1(\mathbf{x}))$ varies in the range $(1, 40)$. Note that the entropy computation is performed using an approximation series summation and for extreme values of the model score, the computation of entropy and thus, uncertainty estimation does become unstable. The primary takeaway is that the score estimation bias depends on relative strengths of priors and evidence, i.e., $\gamma(\mathbf{x})$, which as shown is positively correlated with uncertainty. This observation holds even for the general case where there is differential sampling $\tau > 1$.

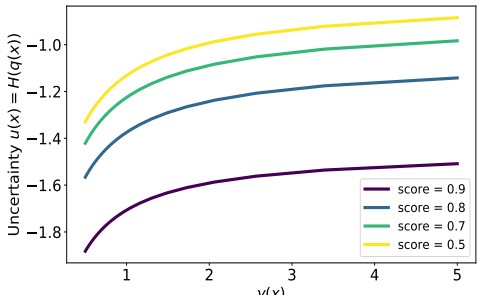

Figure 9: Relation between uncertainty $u(\mathbf{x})(= H(q(x)))$ and $\gamma(\mathbf{x})$ for a fixed score. Here, global prior parameters$(\beta_1^P, \beta_{P)=(5,10)}^0$ and total evidence $(\beta_0(\mathbf{x}) + \beta_1(\mathbf{x}))$ varies in the range $(1, 40)$.

## F  COMPUTATIONAL COMPLEXITY OF DECISION BOUNDARY ALGORITHMS

**Lemma F.1.** *Given a $K \times L$ grid with positive sample counts $[p(i,j)]_{K \times L}$ and total sample counts $[n(i,j)]_{K \times L}$ and any boundary $\mathbf{b} = [b(i)]_{i=1}^K$ that satisfies $precision(\mathbf{b}) \geq \sigma$ and $recall(\mathbf{b}) \geq \eta$, let $b^{chp}(i)$ denote the minimum score threshold $j$ such that $\frac{p(i,j')}{n(i,j')} \geq \sigma$ for all $j' \geq j$, i.e., contiguous high precision region. Then, then the new boundary $\mathbf{b}'$ defined as $b'(i) = min\ (b(i), b^{chp}(i)),\ \forall [i]_1^K$ also satisfies $precision(\mathbf{b}) \geq \sigma$ and $recall(\mathbf{b}) \geq \eta$.*

*Proof.* Let $B'^+$ denote the positive region for the new boundary $\mathbf{b}'$ and $B^{chp}$ the contiguous high precision bins for each uncertainty level, i.e., $B^{chp} = \{(i,j) | j > b^{chp}(i), \forall [i]_1^K, [j]_0^L\}$.

By definition, we have, $B'^+ = \{(i,j) | j > b'(i), \forall [i]_1^K, [j]_0^L\} = B^+ \bigcup B^{chp}$. Given a set of bins $B$, let $P(B)$ and $N(B)$ denote the net positive and total samples within this set of bins. Since $precision(\mathbf{b}) \geq \sigma$, we have $P(B^+) \geq \sigma N(B^+)$. Since $\frac{p(i,j)}{n(i,j)} \geq \sigma, \forall (i,j) \in B^{chp}$, we also note that $P(B) \geq \sigma N(B)$ for any set $B \subseteq B^{chp}$.

Now, the precision for the new boundary is given by

$$
\begin{aligned}
precision(\mathbf{b}') &= \frac{\sum_{(i,j) \in B'^+} p(i,j)}{\sum_{(i,j) \in B'^+} n(i,j)} = \frac{\sum_{(i,j) \in B^+} p(i,j) + \sum_{(i,j) \in B^{chp} \setminus B^+} p(i,j)}{\sum_{(i,j) \in B^+} n(i,j) + \sum_{(i,j) \in B^{chp} \setminus B^+} n(i,j)} \\
&= \frac{P(B^+) + P(B^{chp} \setminus B^+)}{N(B^+) + N(B^{chp} \setminus B^+)} \\
&< \sigma \left( \frac{N(B^+) + N(B^{chp} \setminus B^+)}{N(B^+) + N(B^{chp} \setminus B^+)} \right) \quad \{\text{since}\ (B^{chp} \setminus B^+) \subseteq B^{chp}\} \\
&= \sigma
\end{aligned}
$$

---

[4]$H(q(\mathbf{x})) = \log \mathcal{B}(\alpha_0, \alpha_1) - (\alpha_0 + \alpha_1 - 2)\psi(\alpha_0 + \alpha_1) - \sum_{c \in \mathcal{C}} (\alpha_c - 1)\psi(\alpha_c)$ where $\psi(\cdot)$ is the digamma function and $\mathcal{B}(\cdot, \cdot)$ is the Beta function.

Let $P_0$ denote the total number of positive samples. Then the recall for the new boundary is given by

$$recall(\mathbf{b}') = \frac{\sum_{(i,j) \in B'^+} p(i,j)}{P_0} = \frac{P(B^+) + P(B^{chp} \setminus B^+)}{P_0} \geq \frac{P(B^+)}{P_0} > \eta.$$

Hence, $\mathbf{b}'$ also satisfies the precision and recall bounds. $\qquad\square$

**Theorem F.2. [Restatement of Theorem 5.1]** *The problem of computing the optimal 2D- binned decision boundary (2D-BDB) is NP-hard.*

*Proof.* The result is obtained by demonstrating that any instance of the well-known subset-sum problem defined below can be mapped to a specific instance of a reformulated 2D-BDB problem such that there exists a solution for the subset-sum problem instance if and only if there exists a solution for the equivalent decision boundary problem.

Specifically, we consider the following two problems:

Subset-sum problem: Given a finite set $\mathcal{A} = \{a_1, \ldots, a_t\}$ of $t$ non-negative integers and a target sum $T$, is there a subset $\mathcal{A}'$ of $\mathcal{A}$ such that $\sum_{a_r \in \mathcal{A}'} a_r = T$.

Reformulated 2D-BDB problem: Given a $K \times L$ grid with $p(i,j)$ and $n(i,j)$ denoting the positive and total number of samples for bin $(i,j)$, is there a decision boundary $\mathbf{b} = [b(i)]_{i=1}^K$ such that $precision(\mathbf{b}) \geq \sigma$ and $recall(\mathbf{b}) \geq \eta$.

Let $B^+$ denote the positive region of the boundary, i.e., $B^+ = \{(i,j) | 1 \leq i \leq K, 1 \leq j \leq L; j > b(i)\}$ and $P_0$ denote the total number of positive samples. Then, we require

- $precision(\mathbf{b}) = \frac{\sum_{(i,j) \in B^+} p(i,j)}{\sum_{(i,j) \in B^+} n(i,j)} \geq \sigma$

- $recall(\mathbf{b}) = \frac{\sum_{(i,j) \in B^+} p(i,j)}{P_0} \geq \eta,$

Note that maximizing recall for a precision bound is equivalent to reformulation in terms of the existence of a solution that satisfies the specified precision bound and an arbitrary recall bound.

Given any instance of subset-sum problem with $t$ items , we construct the equivalent decision boundary problem by mapping it to a $(t+1) \times 1$ grid (i.e., $K = t+1, L = 1$ with bins set up as follows.

- $n(i,1) = T; \quad p(i,1) = 2\sigma T,$

- $n(i+1,1) = a_i; \quad p(i+1,1) = 2\epsilon a_i,$

where the parameters $\sigma, \epsilon, \eta$ can be chosen to be any set of values that satisfy

- $0 \leq \sigma \leq \frac{1}{2}, \quad 0 < \epsilon < \frac{\sigma}{2(T+1)}, \quad \eta = \frac{2(\sigma+\epsilon)T}{P_0}.$

We prove that the problems are equivalent in the sense that the solution for one can be constructed from that of the other.

**Part 1: Solution to subset sum $\Rightarrow$ Solution to decision boundary**

Suppose there is a subset $\mathcal{A}'$ such that $\sum_{a_i \in \mathcal{A}'} a_i = T$. Then, consider the boundary $\mathbf{b}$ defined as $b(1) = 1$ and $b(i) = \mathbf{1}[a_i \notin \mathcal{A}']$, i.e., the positive $B^+ = \{(1,1)\} \bigcup \{(i+1,1) | a_i \in \mathcal{A}'\}$. This leads to the following precision and recall estimates.

$$precision(\mathbf{b}) = \frac{\sum_{(i,j) \in B^+} p(i,j)}{\sum_{(i,j) \in B^+} n(i,j)} = \frac{2\sigma T + 2\epsilon \sum_{a_i \in \mathcal{A}'} a_i}{T + \sum_{a_i \in \mathcal{A}'} a_i} = \frac{2(\sigma + \epsilon)T}{2T} \geq \sigma$$

$$recall(\mathbf{b}) = \frac{\sum_{(i,j)\in B^+} p(i,j)}{P_0} = \frac{2\sigma T + 2\epsilon \sum_{a_i\in\mathcal{A}'} a_i}{P_0} = \frac{2(\sigma + \epsilon)T}{P_0} = \eta.$$

Since this choice of $\mathbf{b}$ is a valid boundary satisfying the precision and recall requirements, we have a solution for the decision boundary problem.

**Part 2: Solution to decision boundary $\Rightarrow$ Solution to subset sum**

Let us assume we have a solution for the decision boundary, i.e., we have a boundary $\mathbf{b}$ with $precision(\mathbf{b}) \geq \sigma$ and $recall(\mathbf{b}) \geq \eta$ respectively. Since the positivity rate of the bin $(1,1)$ is $\frac{2\sigma T}{T} = 2\sigma > \sigma$, from Lemma F.1 we observe that the boundary $\mathbf{b}$ is such that $(1,1)$ is in the positive region of the boundary $B^+$.

Consider the subset $\mathcal{A}' = \{a_i | (i+1,1) \in B^+\}$. We will now prove that $\sum_{a_i\in\mathcal{A}'} = T$ which makes it a valid solution for the subset-sum problem.

Suppose that $\sum_{a_i\in\mathcal{A}'} = T' > T$, i.e., $T' \geq T + 1$ since T is an integer. For this case, we have

$$precision(\mathbf{b}) = \frac{\sum_{(i,j)\in B^+} p(i,j)}{\sum_{(i,j)\in B^+} n(i,j)} = \frac{2\sigma T + 2\epsilon \sum_{a_i\in\mathcal{A}'} a_i}{T + \sum_{a_i\in\mathcal{A}'} a_i} = \frac{2\sigma T + 2\epsilon T'}{T + T'}.$$

Since $\epsilon < \frac{\sigma}{2(T+1)}$, we have

$$
\begin{aligned}
precision(\mathbf{b}) &= \frac{2\sigma T + 2\epsilon T'}{T + T'} < \frac{2\sigma T + \frac{2\sigma T'}{2(T+1)}}{T + T'} = \sigma\left(\frac{2T + \frac{T'}{T+1}}{T + T'}\right) = \sigma\left(1 + \frac{T - T' + \frac{T'}{T+1}}{T + T'}\right) \\
&= \sigma\left(1 + \frac{T - \frac{TT'}{T+1}}{T + T'}\right) = \sigma\left(1 - \frac{T(T' - T - 1)}{(T+1)(T + T')}\right) \leq \sigma. \ \{\text{since } T' \geq T + 1\}
\end{aligned}
$$

In other words, $precision(\mathbf{b}) < \sigma$, which is a contradiction since $\mathbf{b}$ is a valid solution to the decision boundary problem.

Next consider the case where $\sum_{a_i\in\mathcal{A}'} = T' < T$. Then, we have,

$$recall(\mathbf{b}) = \frac{\sum_{(i,j)\in B^+} p(i,j)}{P_0} = \frac{2\sigma T + 2\epsilon \sum_{a_i\in\mathcal{A}'} a_i}{P_0} = \frac{2\sigma T + 2\epsilon T'}{P_0} < \frac{2\sigma T + 2\epsilon T}{P_0} = \eta.$$

This again leads to a contradiction since $\mathbf{b}$ is a solution to the decision boundary problem requiring $recall(\mathbf{b}) \geq \eta$. Hence, the only possibility is that $\sum_{a_i\in\mathcal{A}'} = T$, i.e., we have a solution for the subset-sum problem. Since the subset-sum problem is NP-hard (Caprara et al., 2000), from the reduction, it follows that the 2D-BDB problem is also NP-hard. $\square$

## G  DECISION BOUNDARY ALGORITHMS

Here, we provide additional details on the following proposed algorithms from Sec. 5 that are used in our evaluation. These are applicable for both variable or equi-weight binning scenarios.

**Equi Weight DP-based Multi-Threshold algorithm (EW-DPMT)** : We detail the EW-DPMT (Algorithm 1), presented in Sec. 5 here. Let $R(i,m)$, $[i]_1^K$, $[m]_0^{KL}$ denote the maximum true positives for any decision boundary over the sub-grid with uncertainty levels 1 to $i$ and entire score range, such that the boundary has exactly $m$ bins in its positive region. Further, let $b(i,m,:)$ denote the optimal boundary that achieves this maximum with $b(i,m,i')$ denoting the boundary position for the $i'(\leq i)$ uncertainty level. For the base case when $i = 1$, there is a feasible solution only for $0 \leq m \leq L$ which is the one corresponding to $b(1,m,1) = L - m$, since the score threshold index for picking $m$ bins in the positive region will be $L - m$. Now, for the case $i > 1$, we can decompose the estimation of maximum recall as follows. Let $j$ be the number of bins chosen as part of positive region from the $i^{th}$ uncertainty level, then the budget available for the lower $(i - 1)$ uncertainty

levels is exactly $m - j$. Hence, we have, $R(i, m) = \max_{0 \le j \le L} [\pi(i, j) + R(i - 1, m - j)]$, where $\pi(i, j) = \sum_{j'=L-j+1}^{L} p(i, j')$, i.e., the sum of the positive points in the $j$ highest score bins. The optimal boundary $b(i, m, :)$ is obtained by setting $b(i, m, i) = L - j^*$ and the remaining thresholds to that of $b(i - 1, m - j^*, :)$ where $j^*$ is the optimal choice of $j$ in the above recursion.

Performing this computation progressively for all uncertainty levels and positive region bin budgets yields maximum recall over the entire grid for each choice of bin budget. This is equivalent to obtaining the entire PR curve and permits us to pick the optimal solution for a given precision bound. Since the bin-budget can go up to $KL$ and the number of uncertainty levels is $K$, the number of times the maximum recall optimization is invoked is $K^2 L$. The optimization itself explores $L$ choices, each being a $O(1)$ computation since the cumulative sums of positive bins can be computed progressively. Hence, the overall algorithm has $O(K^2 L^2)$ time complexity and $K^2 L$ storage complexity. Algorithm 1 EQUI-WEIGHT DP-BASED MULTI-THRESHOLDS (EW-DPMT) shows steps for computing the optimal 2D-decision boundary. Note that if a solution is required for a specific precision bound $\sigma$, then complexity can be reduced by including all contiguous high score bins with positivity rate $\ge \sigma$ since those will definitely be part of the solution (Lemma F.1).

**Variable Weight DP-based Multi-Threshold algorithm (VW-DPMT)** As discussed in Sec. 5, the general case of the 2D-BDB problem with variable-sized bins is NP-hard, but it permits a pseudo-polynomial solution using a dynamic programming approach. Similar to the equi-weight DP algorithm EW-DPMT, we track the maximum recall solutions of sub-grids up to $i^{th}$ uncertainty level with a budget over the number of positive samples.

Let $R^{var}(i, m)$ denote the maximum true positives for any decision boundary over the sub-grid with uncertainty levels $1$ to $i$ and the entire score range such that the boundary has exactly $m$ samples in its positive region. We can then use the decomposition,

$$R^{var}(i, m) = \max_{0 \le j \le L} [\pi(j) + R^{var}(i - 1, m - \nu(j))],$$

where $\pi(i, j) = \sum_{j'=L-j+1}^{L} p(i, j')$ and $\nu(i, j) = \sum_{j'=L-j+1}^{L} n(i, j')$.

Algorithm 4 provides details of the implementation assuming a dense representation for the matrix $R^{var}$ (Eqn. G) that tracks all the maximum true positive (i.e., unnormalized recall) solutions for sub-grids up to different uncertainty levels and with a budget on the number of samples assigned to the positive region. For our experiments, we implemented the algorithm using a sparse representation for $R^{var}$ that only tracks the feasible solutions.

**Greedy Multi-Thresholds (GMT)** Algorithm 2 provides the details of this greedy approach where we independently choose the score threshold for each uncertainty level. Since all the score bin thresholds are progressively evaluated for each uncertainty level, the computational time complexity is $O(KL)$ and the storage complexity is just $O(K)$. However, this approach can even be inferior to the traditional approach of picking a single global threshold on the score, which is the case corresponding to a single uncertainty level. ST algorithm can be viewed as a special case of GMT algorithm where only one uncertainty level is considered (i.e. $K = 1$).

**Multi Isotonic regression Single Threshold (MIST)** As mentioned earlier, the isotonic regression-based approach involves performing isotonic regression (Barlow & Brunk, 1972) on each uncertainty level to get calibrated scores that are monotonic with respect to the score bin index. Bins across the entire grid are then sorted based on the calibrated scores and a global threshold on the calibrated score that maximizes recall while satisfying the desired precision bound is picked. In our implementation, we use the isotonic regression implementation is `scikit-learn`, which has linear time in terms of the input size for $L_2$ loss (Stout, 2013). Since the sorting based on calibrated scores is the most time consuming part, this algorithm has a time complexity of $O(KL \log(KL))$ and a storage complexity of $O(KL)$. For our experiments, we performed isotonic regression for each of the $K$ uncertainty levels directly using the samples instead of the aggregates at $L$ score bins. When the $K$ uncertainty bins are equi-weight, this is essentially the case where $L = N/K$.

---

**Algorithm 2** Greedy Decision Boundary - Multiple Score Thresholds [GMT]

---

**Input:** Variable-sized $K \times L$ grid with positive sample counts $[p(i,j)]_{K \times L}$ and total sample counts $[n(i,j)]_{K \times L}$, overall sample count $N$, precision bound $\sigma$.

**Output:** (unnormalized) recall $R^*$ and corresponding boundary $\mathbf{b}^*$ for precision $\geq \sigma$ with greedy approach.

**Method:**

    *// Pre-computation of cumulative sums of positives*

    **for** $i = 1$ to $K$ **do**

        $\pi(i,0) = 0$

        $\nu(i,0) = 0$

        **for** $j = 1$ to $L$ **do**

            $\pi(i,j) = \pi(i,j-1) + p(i, L-j+1)$

            $\nu(i,j) = \nu(i,j-1) + n(i, L-j+1)$

        **end for**

    **end for**

    *// Initialization*

    $R = 0$

    *// Independent Greedy Score Thresholds*

    **for** $i = 1$ to $K$ **do**

        $j^* = \underset{0 \leq j \leq L, \ s.t. \frac{\pi(i,j)}{\nu(i,j)} \geq \sigma}{\operatorname{argmax}} [\pi(i,j)]$

        $b(i) = j^*$

        $R = R + \pi(i, j^*)$

    **end for**

    $R^* = R, \ \mathbf{b}^* = b(:)$

    **return** $(R^*, \mathbf{b}^*)$

---

---

**Algorithm 3** Greedy Decision Boundary - Global Threshold on Score Recalibrated with Isotonic Regression [MIST]

---

**Input:** Variable-sized $K \times L$ grid with positive sample counts $[p(i,j)]_{K \times L}$ and total sample counts $[n(i,j)]_{K \times L}$, overall sample count $N$, precision bound $\sigma$.

**Output:** (unnormalized) recall $R^*$ and corresponding boundary $\mathbf{b}^*$ for precision $\geq \sigma$ with greedy approach.

**Method:**

  *// Recalibrate each row using isotonic regression*

  **for** $i = 1$ to $K$ **do**

    $[s^{iso}(i,j)]_{j=1}^{L} = \text{IsotonicRegression}([(p(i,j), n(i,j))]_{j=1}^{L})$

  **end for**

  *// Get a global threshold on calibrated score*

  *// rank is descending order 0 to maxrank - low rank means high positivity*

  $[rank(i,j)]_{K \times L} = \text{Sort}([s^{iso}(i,j)]_{K \times L})$

  $maxrank = \max\limits_{[i]_1^K \ [j]_1^L} rank(i,j)$

  $\pi(0) = 0, \nu(0) = 0$

  $r = 0$

  **repeat**

    $r = r + 1$

    $\pi(r) = \pi(r-1) + \sum_{(i,j)|\ rank(i,j)=r} [s^{iso}(i,j)n(i,j)]$

    $\nu(r) = \nu(r-1) + \sum_{(i,j)|\ rank(i,j)=r} [n(i,j)]$

  **until** $\left( (\frac{\pi(r)}{\nu(r)} < \sigma) \vee (r > maxrank) \right)$

  $r^* = r - 1$

  *// Obtain score thresholds for different uncertainty levels*

  $R = 0$

  **for** $i = 1$ to $K$ **do**

    **if** $\{j|rank(i,j) \geq r^*\} = \emptyset$ **then**

      $j^* = L$

    **else**

      $j^* = \underset{j|rank(i,j) \geq r^*}{\arg\min} [j]$

    **end if**

    $b(i) = j^*$

    **for** $j = j^* + 1$ to $L$ **do**

      $R = R + p(i,j)$

    **end for**

  **end for**

  $R^* = R, \ \mathbf{b}^* = b(:)$

  **return** $(R^*, \mathbf{b}^*)$

---

---

**Algorithm 4** Optimal Decision Boundary for Variable-Weight Bins [VW-DPMT]

---

**Input:** Variable-sized $K \times L$ grid with positive sample counts $[p(i,j)]_{K \times L}$ and total sample counts $[n(i,j)]_{K \times L}$, overall sample count $N$, precision bound $\sigma$.

**Output:** maximum (unnormalized) recall $R^*$ and corresponding optimal boundary $\mathbf{b}^*$ for precision $\geq \sigma$.

**Method:**

  *// Initialization*
  $R(i,m) = -\infty$; $b(i,m,i') = -1$; $[i]_1^K$, $[i']_1^K$, $[m]_0^N$)
  *// Pre-computation of cumulative sums of positives*
  **for** $i = 1$ to $K$ **do**
    $\pi(i,0) = 0$
    $\nu(i,0) = 0$
    **for** $j = 1$ to $L$ **do**
      $\pi(i,j) = \pi(i,j-1) + p(i,L-j+1)$
      $\nu(i,j) = \nu(i,j-1) + n(i,L-j+1)$
    **end for**
  **end for**
  *// Base Case: First Uncertainty Level*
  **for** $j = 0$ to $L$ **do**
    $m = \nu(1,j)$
    $R(1,m) = \pi(1,j)$
    $b(1,m,1) = L - j$
  **end for**
  *// Decomposition: Higher Uncertainty Levels*
  **for** $i = 2$ to $K$ **do**
    **for** $m = 0$ to $\sum_{i'=0}^{i} N^{cum(i,j)}$ **do**
      $j^* = \underset{0 \leq j \leq L}{\operatorname{argmax}}[\pi(i,j) + R(i-1, m - \nu(i,j))]$
      $R(i,m) = \pi(i,j^*) + R(i-1, m - \nu(i,j^*))$
      $b(i,m,:) = b(i-1, m - \nu(i,j^*), :)$
      $b(i,m,i) = L - j^*$
    **end for**
  **end for**
  *// Maximum Recall for Precision*
  $m^* = \underset{0 \leq m \leq KL \ s.t. \frac{R(K,m)}{m} \geq \sigma}{\operatorname{argmax}}[R(K,m)]$
  $R^* = R(K,m^*)$; $\mathbf{b}^* = b(K,m^*,:)$
  **return** $(R^*, \mathbf{b}^*)$

---

# H  EXTENSION OF RESULTS TO OTHER SETTINGS

We briefly discuss the possibility of extending our results to broader settings.

## H.1  GENERAL PRIORS AND UNCERTAINTY MODELING.

The primary objective of the analysis in Sec. 3 was to demonstrate that there is a systemic dependence of model score estimation bias on uncertainty for common scenarios. In our current work, we characterise this relationship (Theorems 3.1 and E.4 ) for the case where (i) the data is generated as per Fig. 2 with global Beta distribution prior, and (ii) the estimation of model score and uncertainty is performed using Posterior Network.

Note that the first part of the result, i.e., the relationship between the expectation of test and true conditional class probabilities conditioned on the train positive class probability holds true independent of (ii). The second part of the result, which is the relationship with model score depends on the choice of uncertainty modeling approach (ii).

When the first condition(i) does not hold, i.e., it becomes intractable to obtain an equivalent closed form result. However, for a global unimodal prior (not necessarily a Beta distribution), the

expectation of test and true conditional class probabilities conditioned on the train positive class probability is still shifted away from the train distribution in the direction of the global prior since the posterior distribution of the true positivity conditioned on the train positivity combines the effects of the evidence (train positivity) and the global prior.

## H.2 MULTI-CLASS CLASSIFICATION.

In our current work, we focused on binary classification . However, the analysis in Sec. 3 on the relationship between model score estimation bias and uncertainty (Theorems 3.1 and E.4 ) readily generalize to a multi-class classification with more than two classes when the data generation in Figure 2 uses a global Dirichlet prior instead of Beta prior with a fixed re-sampling policy. For such scenario, one can derive the expectation of true and test conditional class probabilities conditioned on the train conditional class probabilities using Bayes law. As before, when the modeling and uncertainty estimation is based on Posterior Networks, we can obtain the relationship with respect to the model's conditional class probabilities. Intuitively, the true and test conditional class probability distribution diverges from the corresponding train distribution in the direction of the global prior with the magnitude of divergence determined by the strength of evidence, which correlates with models' epistemic uncertainty.

On the other hand, generalization of the decision-making strategy to more than two classes is somewhat non-trivial since the notion of decision boundary becomes more complex in the case of more than two classes and requires characterization of the objective function that the decision boundaries are meant to optimize. One possible formulation that suits many real-world applications is to define the decision boundaries so as to maximize micro-average recall with an overall or class-wise precision bounds. Monotone operators over the conditional probability distributions that also combine uncertainty-based discriminating signals is a possible research direction to explore. Previous works based on set predictions (Mortier et al., 2021; Angelopoulos et al., 2021) also attempt to address this problem scenario but without explicit use of uncertainty estimates.

# I  NOTATIONS

Table 8: Notations used within the paper and their definitions.

| Symbol | Definition |
| --- | --- |
| $\mathbf{x}$ | an input instance or region |
| $y$ | target label for an input sample $\mathbf{x}$ |
| $\mathcal{C}$ | set of class labels $\{0, 1\}$ |
| $c$ | index over the labels in $\mathcal{C}$ |
| $[i]_{lb}^{ub}$ | index iterating over integers in $\{lb, \cdots, ub\}$ |
| *Estimation Bias and Posterior Network Related* | |
| $\mathbf{P}(\cdot)$ | Probability distribution |
| $q(\mathbf{x})$ | Distribution over class posterior at $\mathbf{x}$ output by Posterior Network |
| $H(q(\mathbf{x}))$ | differential entropy of distribution $q(\mathbf{x})$ |
| $\text{Beta}(\alpha_0, \alpha_1)$ | Beta distribution with parameters $\alpha_0, \alpha_1$ $\alpha_c(\mathbf{x})$ |
| Parameters of Beta distribution $q(\mathbf{x})$ for class $\mathcal{C}$ | |
| $\beta_c^P$ | Parameters of Model prior for class $\mathcal{C}$ |
| $\beta_c^T$ | Parameters of True prior for class $\mathcal{C}$ |
| $\beta_c(\mathbf{x})$ | Pseudo counts for class $\mathcal{C}$ |
| $N_c$ | observed counts for class $\mathcal{C}$ |
| $\mathbf{z}(\mathbf{x})$ | penultimate layer representation from the model |
| $\phi$ | parameters of normalizing flow in Posterior Networks |
| $u(\mathbf{x})$ | Uncertainty for $\mathbf{x}$ |
| $S^{model}(\mathbf{x}), \ s(\mathbf{x})$ | Model score for positive class |
| $S^{true}(\mathbf{x})$ | true positivity in input region $\mathbf{x}$ |
| $S^{train}(\mathbf{x})$ | empirical positivity in the train set for input region $\mathbf{x}$ |
| $S^{test}(\mathbf{x})$ | empirical positivity in the test set for input region $\mathbf{x}$ |
| $\tau$ | differential sampling rate for negaatives |
| $n = n(\mathbf{x})$ | evidence at input region $\mathbf{x}$ given by $\beta_1(\mathbf{x}) + \beta_2(\mathbf{x})$ |
| $\xi$ | positive class fraction in global prior |
| $\omega$ | positive class fraction in model prior |
| $\lambda(\mathbf{x})$ | ratio of global priors to evidence |
| $\gamma(\mathbf{x})$ | ratio of model priors to evidence |
| $\nu$ | ratio of global priors to model priors |
| $Q(\cdot)$ | distribution of true positivity conditioned on a fixed train positivity rate |
| *Decision Boundary Related* | |
| $D^{train}, D^{hold}, D^{test}$ | Data split for training the model, calibrating the decision boundary and testing |
| $\mathbf{b}$ | decision boundary defined in terms of score and uncertainty thresholds |
| $\psi_{\mathbf{b}}(\mathbf{x})$ | labeling where samples that satisfy boundary thresholds are positives |
| $\mathbf{b}(u)$ | decision boundary specified by a score threshold for a fixed uncertainty level |
| $\mathcal{S}$ | Range of score values |
| $\mathcal{U}$ | Range of uncertainty values |
| $K$ | Number of uncertainty bins |
| $L$ | Number of score bins |
| $\rho(s, u) = (\rho^{\mathcal{S}}(s), \rho^{\mathcal{U}}(u))$ | Partitioning function that maps score and uncertainty values to bin-index $(i, j)$ |
| | Continued on next page |

**Table 8 – continued from previous page**

| Symbol | Definition |
|---|---|
| $R(i, m)$ | max. true positives for any boundary upto the $i^{th}$ uncertainty level with exactly $m$ positive bins in EW-DPMT |
| $b(i, m, :)$ | max. recall boundary for the sub-grid upto uncertainty level $i$, with exactly $m$ positive bins |
| $p(i, j)$ | count of positives in the $(i, j)$th bin |
| $n(i, j)$ | count of samples in the $(i, j)$th bin |
| $\pi(i, j)$ | count of positive samples in the $j$ highest score bins for uncertainty level $i$ |

