# OpenReview forum: "Leveraging Uncertainty Estimates To Improve Classifier Performance"
_ICLR.cc/2024/Conference — ICLR 2024 poster_

### Official Review · Reviewer_D6i1 · 2023-10-19

**Soundness:** 1 poor
**Presentation:** 1 poor
**Contribution:** 1 poor
**Rating:** 3
**Confidence:** 4

**Summary:**

This paper tries to improve classification performance by leveraging uncertainty estimates. To this end, the authors focus on posterior networks, a novel class of classification models that has been recently introduced as an alternative for Bayesian neural networks in OOD detection. Posterior networks model a second-order distribution that is parameterized by a neural network using specific loss functions. In this paper, the authors focus on binary classification, and the considered second-order distribution is a Beta distribution.

In addition, the authors assume that the data generation process also has a Beta prior over the Bernoulli parameter of the first-order distribution. Within this framework, the authors analyze model estimation bias and they show that it depends both on the aleatoric and epistemic uncertainty. Furthermore, they formulate decision boundary selection in terms of both types of uncertainty, and they prove that this problem is NP hard.

**Strengths:**

Uncertainty quantification is a hot topic in machine learning. The submission also contains material that is novel.

**Weaknesses:**

This paper is not well written. I have the impression that the authors lack a thorough understanding of the literature. They use unconventional notions, and they don't discuss the problem in a formal way. Let me make this point clear.

Introduction:
- the authors write "benefits of combining model score with estimates of aleatoric and epistemic uncertainty". This is a very weird reasoning, because the model score corresponds to the estimate of aleatoric uncertainty. (btw, better to use conditional class probabilities instead of model score, because model score is a very generic term)
- RQ1: Does model score estimation bias (deviation from positivity) depend on uncertainty? This is a really weird sentence that is difficult to understand. I only understood that sentence after reading Section 3. You first have to explain what you mean with model score estimation bias. Is it a bias in the sense of the traditional bias-variance decomposition or is it something else? The same with "positivity". That's a term that is never used in the literature. What you mean is the probability of the positive class on population level. After reading Section 3 it became clear to me what you mean with "uncertainty". It is the differential entropy of the second-order distribution. I would argue that this is a way to measure epistemic uncertainty with posterior networks, but you have to be more precise in your description.
- Same comment for "test positivity rate". Never heard of this term before. Please define it.

Related work:
- The overview of related work is described in a very vague way. What you call "Uncertainty modelling" are methods that intend to quantify aleatoric and  epistemic uncertainty, such as Bayesian methods, ensemble methods and evidential deep learning methods.
- Please also describe methods that only quantify aleatoric uncertainty, such as traditional probabilistic models. In that context, I would argue that "calibration metrics" are a way to assess whether standard probabilistic methods quantify aleatoric uncertainty correctly.

Section 3:
- Posterior networks: please motivate why this group of methods is your first choice to improve classification performance by leveraging epistemic uncertainty scores. Recent papers have critisized this group of methods for not quantifying epistemic uncertainty in a correct manner, see e.g. Bengs et al. Pitfalls of epistemic uncertainty quantification through loss minimization, Neurips 2022. So, I would argue that it is weird to consider this group of methods as first choice.
- 3.1: the uncertainty score u(x) is a measure of epistemic uncertainty. Please be more precise.
- 3.2: notions such as "true positivity" and "empirical positivity in the train and test set" are very weird terms. I assume that you refer to true and estimated conditional class probabilities for the positive class. Please be more precise.
- 3.2: Why is the true positivity sampled from a Beta distribution? Is this to generate synthetic data, or what is the purpose of this? Since you are presenting a theoretical result in Theorem 3.1, I assume that you have a different purpose of considering a Beta distribution.
-Theorem 3.1: "When data is generated according to Figure 2". Unclear what this means. The figure is not detailed enough. The data generation process has to be described in a formal way.

Because of the very imprecise write-up, it is difficult to follow what the novelty of this paper is. I have to admit that I was completely lost after Theorem 3.1, even though I am very familiar with posterior networks.

**Questions:**

See above.

---

> ### Author Response · Authors · 2023-11-20
> **Rebuttal To Reviewer D6i1 [1/2]**
>
> We are grateful to the reviewer for their valuable feedback. We agree with the reviewer that we could have done a better job at paper presentation. In particular, we should have defined all new terms the first time they are introduced than at a later point. We will do our best to incorporate all the inputs as we revise the draft. Please find our response below
>
> We feel the reviewer has completely missed out on the main contributions of the paper, which are not necessarily tied to the choice of using Posterior networks to quantify uncertainty. To clarify, we make two theoretical contributions. Note that the primary objective of the first contribution is to motivate the second one.
>
> 1) First, we try to establish that that there exist common scenarios where empirically observed conditional probability $p(y=1|x)$ of the positive class in the training dataset as well as the model estimate deviates from that of the underlying distribution and the test set in a systemic way with dependence on the density of observations at x (which is correlated to “uncertainty”) (Theorem 3.1 - Specifically the common scenario we consider is one where the data is generated as per Fig 2 with the actual generative distribution mentioned in Sec 3.2 as well as Appendix E.1)
>
> 2) Second, we provide algorithms to combine the model estimated $p(y=1|x)$ and ANY other discriminating signal (such as uncertainty estimates) to arrive at a decision boundary that better optimizes recall at a precision bound.
>
> As we mention in footnote 2 (Pg2), Theorem 3.1 (a) on deviation between $p(y=1|x)$ in the training data and that of test data or the underlying distribution is completely independent of the choice of uncertainty modeling method and is not dependent on Posterior Networks. The same is true for Sections 4 & 5.
>
> However, Theorem 3.1(b) characterizes how Theorem 3.1(a) applies for the case where the the model score and uncertainty are obtained using Posterior Networks. The primary reason we included this result is because most of our experiments are based on Posterior Networks and we wanted to compare the empirically observed trends with the theoretically expected behavior. We will rewrite Sec 3 to clarify this aspect.
>
>
> **Responses to Detailed Comments:**
>
> **Comment 1 :** This paper is not well written ... Let me make this point clear.
> **Response:**  (a) We acknowledge that we used some non-standard terms such as train/test positivity and model score estimation bias. We will revise the draft to ensure these are defined precisely the very first time they are used rather than in footnotes or further down in the writeup or appendix. (b) We do present a formal definition of the decision-making problem in Sec 4. Could the reviewer kindly clarify their expectations on the problem definition?
>
> **Comment 2:** **Introduction:** the authors write "benefits ... very generic term)
> **Response:**
> (a) Yes, aleatoric uncertainty is a deterministic function of model score but that is not true of epstemic uncertainty and will rephrase the sentence to read better.
> (b) We are using the phrase “model score“ instead of ”conditional probability of the positive class as estimated by model“ only for brevity and have clarified this in Footnote 1[Pg 1]
>
> **Comment 3:** RQ1: Does model score estimation bias ...(c) Same comment for "test positivity rate". Never heard of this term before. Please define it.
> **Response:**  (a) We do mention that the “model score estimation bias“ is the “difference between test positivity rate and the model score varies with uncertainty” [Pg 2 Para 2] but will move this up to where RQ1 is mentioned. (b) Apologies for not having defining “positivity rate” upfront. It is just the “probability of the positive class on population level”. “Train positivity” and “Test positivity” corresponds to the training and test distribution respectively. We will make it a point to clarify this as well upfront as we revise the draft. (c) Just to clarify in case of Posterior networks and Theorem 3.1(b), “uncertainty” refers to the differential entropy of the second-order distribution in Posterior networks. However for Sections 4 and 5, we are using the term “uncertainty” in a broader sense and have also included results using other methods for estimating uncertainty (e.g. MCDropout) [Sec 6.2, Pg 8, RQ2 and Appendix C.5, Table 5].
>
> **Comment 4: Related work:** The overview ... aleatoric uncertainty correctly.
> **Response:**  We will expand the related work section to have a more detailed discussion of various uncertainty modeling methods including those focusing on aleatoric uncertainty as well. We focused on methods that also capture epistemic uncertainty because given the model score $p(y=1|x)$, epistemic uncertainty provides additional discriminating ability to determine the optimal decision boundary unlike aleatoric uncertainty that has a deterministic relationship with model score and no additional information.

---

> ### Author Response · Authors · 2023-11-20
> **Rebuttal To Reviewer D6i1 [2/2]**
>
> **Comment 5: Section 3:**  Posterior networks: ... consider this group of methods as first choice.
> **Response:**  First, we wish to clarify that our work does not qualify any uncertainty quantifying/estimating method (eg: Posterior / MC-dropout) as being superior over the other. We are aware of the work by Bengs et al. which discusses why optimizing the model based on uncertain-cross entropy loss function does not provide faithful estimates of epistemic uncertainty and will include it in the revised version of related work, but the gaps in the optimization approach are not central to our work.
>
> Our choice of Posterior Networks for our experiments was based on two considerations. First, we already had an existing implementation for one of our real-world applications and experiments using 2D-decision boundary algorithms yielded promising results. Second, for certain controlled scenarios (Fig 2), we could derive a closed analytic form for the model score estimation bias in terms of the model score, ratios of priors and observed evidence ($\gamma(x)$ ), which is positively correlated with the notion of “uncertainty” (entropy of the epistemic distribution) in Posterior networks, allowing us to compare empirical observations with expected behavior.
>
> Note that it is quite possible that other methods of quantifying epistemic uncertainty yield superior results in terms of optimizing decision boundaries and that remains an area of further exploration. We do provide results on MC-Dropout method as well [Sec 6.2, Pg 8, RQ2 and Appendix C.5, Table 5] and observe similar benefits in combining  uncertainty estimates with conditional class probabilities to oprimize recall at a precision bound. We will attempt to include results from other recent approaches in a revised version of the Appendix.
>
>
>
> **Comment 6: Section 3.1** the uncertainty score $u(x)$ is a measure of epistemic uncertainty. Please be more precise.
> **Response:**  The uncertainty score $u(x)$ needs to definitely capture epistemic uncertainty since that is the element that provides additional information beyond the model estimated $p(Y=1|x)$, but depending on the modeling method, it could also have a dependence on the aleatoric uncertainty. We will clarify this in the revision.
>
> **Comment 7: Section 3.2**  notions such as "true positivity" ... Please be more precise.
> **Response:**  We are using these terms only for brevity and will make it a point to define these first.
>
> **Comment 8: Section 3.2:** (a) Why is the true positivity ... The data generation process has to be described in a formal way.
> **Response:** (a) We should have explained our choice of data generation setting better. Before going to Sec 4-5, we wanted to first establish that there is a systemic deviation between the conditional probability $p(y=1|x)$ from the model and the observed distribution on the test set and that the deviation has a dependence on “uncertainty”. Since the general case was not as tractable, we restricted our analysis to a common representative scenario where we have Beta prior and differential sampling across classes for train and test set. Being able to verify the theoretical analysis with synthetic data was a secondary motivation. (b) We will update Figure 2 to include the exact distributions mentioned in Sec 3.2 and also Appendix E.1. We, in fact, had a more detailed figure but chose the simpler one for lack of space.
>
>
> **Comment 9:**  Because of the very imprecise ... very familiar with posterior networks.
> **Response:** Thanks again for your valuable feedback and taking the time to read through our submission. We hope we have answered some of your questions. Based on the clarifications, if possible, please do take a look at Section 4-6 to share any further feedback or revise your score.

---

### Official Review · Reviewer_i837 · 2023-10-26

**Soundness:** 3 good
**Presentation:** 2 fair
**Contribution:** 2 fair
**Rating:** 6
**Confidence:** 4

**Summary:**

The submission addresses making decisions under uncertainty. More precisely, it considers the binary classification setting, where an instance must be classified into one of two classes, and where the classifier provides a classification score with an associated level (score) of uncertainty.

After a succinct reminder of previous works related to uncertainty modelling and decision-making under uncertainty, the paper first adopts a Bayesian view, and shows that the test positivity rate depends on both the positivity (classification) and uncertainty scores. Then, it proposes to transform the classification problem into picking a class based on both scores, by partitioning the corresponding 2D space (model x uncertainty scores) into bins, thus leading to a 2D decision boundary (instead of a 1D decision threshold for the model score) which maximizes recall for a desired precision level. The paper mentions several strategies for constructing the bins, and then proceeds with an empirical evaluation of the proposed strategy.

**Strengths:**

The paper addresses an interesting and important problem—making decisions under uncertainty. It covers the topic in a rather complete way, by first highlighting the relationship between uncertainty and estimation bias, and then proposing a strategy for improving decision-making by taking the uncertainty level into account.

The proposal is overall sound, with some theoretical grounding as well as an experimental study so as to support the claims.

**Weaknesses:**

My main criticism is that the paper ignores a large part of the literature on decision-making under uncertainty. There already exist many works dedicated to this topic, with the purpose of taking uncertainty into account to improve the decision-making process or abstaining to make decisions when uncertainty is too high. Some of these works are rooted in formalisms alternative to probabilities, while others make use of the classical probabilistic framework. It is obviously not possible to mention all of these works here—see e.g. cautious, imprecise or uncertain classification; I'll only mention several of them here:

[1] Mathias C. M. Troffaes. Decision making under uncertainty using imprecise probabilities. International Journal of Approximate Reasoning, Volume 45, Issue 1, May 2007, Pages 17-29.

[2] Thierry Denoeux. Decision-making with belief functions: A review. International Journal of Approximate Reasoning, Volume 109, June 2019, Pages 87-110.

[3] Thomas Mortier, Marek Wydmuch, Krzysztof Dembczyński, Eyke Hüllermeier, Willem Waegeman. Efficient set-valued prediction in multi-class classification. Data Mining and Knowledge Discovery, Volume 35, Issue 4, Jul 2021, Pages 1435-1469.

Besides, it is difficult to see to which extent the analyzes reported in the proposal can be generalized outside of the framework considered here, either regarding the analysis of estimation bias (Section 3.2) or the proposed decision-making strategy (Sections 4 and 5). As stated by the authors, the Bayesian framework considered here may likely not hold for all datasets. The analysis but also the proposed decision-making strategy seem hard to generalize to more than two classes (in the latter case, for computational reasons). It would have been nice to discuss these limitations in the paper.

The writing should be improved: there are typos; the general structure of the paper might be improved (e.g., Sections 4 "2-D Decision Boundary Problem" and 5 "2-D Decision boundary algorithms" could be agregated); some parts may be developed for the sake of clarity (e.g., the "Dynamic Programming (DP) Algorithm" part in Section 5.2, which is quite verbose but lacks formalization). I do not understand why the numbering of subsections changes from a bold, small-letter font in Section 3 (see e.g. Sections 3.1 and 3.2 page 3) to a capital font in Section 5 (e.g. Sections 5.1, 5.2 and 5.3 pages 6-7). Figures are hard to read (very small); Equation 2 is displayed in an awkward manner; Algorithm 1 is a bit packed and consequently difficult to read.

Typos (non-exhaustive list):
- "relative efficacy" (page 2),
- "Under what settings" (page 2),
- "via Bernoulli distribution" (page 3),
- "In the case of train set" (page 3),
- "corresponds to oversampled negative class" (page 3),
- "expected true and test positivity rate" (page 4),
- "$b(u)$ is score threshold" (page 5),
- "upto" (page 6),
- "feasible solution exists" (page 6),
- "bayesian approximation" (page 10, references),
etc.

**Questions:**

Why choosing the differential entropy of distribution $H(q(\mathbf{x}))$ as a measure of uncertainty ? On one hand, this intuitively makes sense, but it also has been criticized by some authors (see e.g. the works of Hüllermeier).

The interpretation of $\gamma(\mathbf{x})$ in Section 3.2 is not really an interpretation. Since the fact that $u(\mathbf{x})$ is correlated with $\gamma(\mathbf{x})$ is central in the score estimation bias depending on score and uncertainty, this part may be further developed and discussed.

Can you provide insights on generalizing the analysis of estimation bias to other settings (as compared to Fig. 2) ?

The decision-making approach seems debatable. In Section 4: is it reasonable to leave b unconstrained ? Shouldn't a monotonicity assumption be imposed ? Is it reasonable to always keep the decision-making process precise ? Wouldn't it be preferable to abstain in presence of a high (epistemic) uncertainty ? Aleatoric and epistemic uncertainties are here not distinguished from each other, which seems to be an issue: what if $u(\mathbf{x})$ is maximal ? Is it reasonable to make decisions in this case ? What is the effect of discretizing the Score $\times$ Uncertainty space ? To this extent, shouldn't the independent splitting on both dimensions avoided ?

The "proof" that the partitioning problem of the Score $\times$ Uncertainty space is NP-hard (mentioned in the introduction) seems to directly proceed from the bin-packing problem behind.

Equi-weight binning case, "it is possible that a bin with lower positivity rate might be preferable to one with higher positivity rate due to different number of samples": can you be a bit more specific ?

Binning strategy: I understand that for any given score level, the uncertainty bin indices are monotonic wrt actual values; however, this does not correspond to the same level of model score. Des this have an impact on the results ? More generally, what is the impact of the binning strategy on the overall procedure ? This latter may be sensitive to the bins computed.

---

> ### Author Response · Authors · 2023-11-21
> **Response to Reviewer i837 [1/4]**
>
> We are sincerely grateful to the reviewer for taking the time to read through the entire draft and sharing their extremely valuable feedback. Please find our response below
>
>
> **Comment 1 :** My main criticism is that the paper ignores a ... when uncertainty is too high. .... “
>
> **Response:**  Thanks for the pointers. We acknowledge this gap and will definitely expand the related work to include a discussion of the suggested papers . While the papers do deal with decision making under uncertainty, the problem setting and contributions are slightly different as we discuss below.
>
>
> 1. Decision-Making with Belief Functions (Deneoux et al. , Troffaes et al. ) : These works deal with decision making where an input state or policy (possibly multi-dimensional) maps to a consequence which results in a real-valued payoff function with uncertainty modeled via (possibly Bayesian) belief functions. While there are parallels to our decision boundary estimation, in these works, the decision making over competing options is based on a single risk-adjusted (or uncertainty adjusted) utility determined by various axiomatic criterion, e.g., minmax, Hurwicz without any data-based calibration. Second, most of this work is applicable for utility functions that are additive over elements of the input decision space, e.g., overall utility of a stock portfolio can be expressed as the sum of utilities over the individual stocks.. Lastly, there is no discussion around bias in estimating model scores or the effects of differential sampling.
> 2. Set-value prediction: Mortier et al. present an efficient approach for set-value predictions in a multi-class setting based on maximizing the expected utility. This work differs from ours in three key aspects. First, the focus of Mortier et al. is on efficient optimization over the power set of classes with the gains being primarily relevant for multi-class settings with a large number of classes where the set of possible prediction sets, (i.e., any non-empty element of the power set) is extremely large. For the binary classification scenario, there are only 3 prediction sets { {0}, {0,1}, {1} }. Secondly, they do not consider the question of the model estimated conditional probabilities being biased which is an important focus of our work. Lastly, while Mortier et al., discuss unconstrained optimization of multiple different utility functions, such as aggregated precision, recall and f-measure, they do not explicitly consider optimizing recall for a precision bound or vice-versa which is more common in real-world applications and the objective function that we focus on.
>
> **Comment 2 :** Besides, it is difficult to see ... nice to discuss these limitations in the paper.
>
> **Response:**  The decision making strategy in Sec 4-5 is independent of the Bayesian framework and is applicable for datasets following any distribution as well as any choice of uncertainty modeling method.
>
> The purpose of Sec 3 was primarily to motivate the 2D decision-making problem by establishing that that there exist some common scenarios where the conditional probability $p(y=1|x)$ of the positive class in the training dataset as well as the model estimate deviates from that of the underlying distribution and the test set in a systemic way with dependence on uncertainty. In Sec 4-5, we do NOT make any assumption about the nature of the dependence on the uncertainty.
>
> (b) The analysis in Sec 3 [both Theorem 3.1 (a) and (b)] does in fact generalize to multi-class setting with more than two classes when we consider a global Dirichlet prior instead of a Beta prior. We can express the expectation of true and test conditional class probabilities conditioned on the train conditional class probabilities. In case of Posterior networks, we can do the same while conditioning on model’s conditional class probabilities.
>
> On the other hand, generalization of the decision-making strategy to more than two classes is somewhat non-trivial since it first requires characterizing the objective function that the decision boundary(ies) need to be optimized for . We are exploring this setting by considering monotone operators over the conditional probability distributions that also take into account other uncertainty-based discriminating signals. Having said that, there are a large number of real-world applications that require decision making over binary classes which ensures broad applicability. We also consider differential sampling which is a common to multiple large scale applications.
>
> **Comment 3:**  The writing should be improved: ... difficult to read.
> Typos (non-exhaustive list): ... “
>
> **Response:**  Thanks so much for these suggestions. We will address all of these and do a more careful proof reading as we revise our draft.

---

> ### Author Response · Authors · 2023-11-21
> **Response to Reviewer i837 [2/4]**
>
> **Comment 4 : Q1:** Why choosing the differential entropy ... authors (see e.g. the works of Hüllermeier).
>
> **Response:**  We would like to clarify that the decision-making strategy in Sec 4-5 does not assume anything about the nature of uncertainty modeling. In Theorem 3.1(b) and for our experimental section, we focus on Posterior networks and choose the differential entropy of the epistemic distribution as a measure of uncertainty. The reason for this is that it is correlated to density of observations and thus, the ratio of prior to evidence (\lambda(x), which impacts the expected test positivity rate conditioned on the train positivity rate.
>
> We are aware of the work by Bengs, Hullermeier et al . which discusses why optimizing the model based on uncertain-cross entropy loss function does not provide faithful estimates of epistemic uncertainty but the gaps in the optimization approach are not central to our work.
>
> We are only focused on optimizing the decision boundary and do not qualify any uncertainty quantifying/estimating method as being superior over the other in terms of estimation of “epistemic uncertainty” or better calibrated estimates. Do note that the lack of calibration of uncertainty estimates does not also impact the decision boundary estimation.
>
> Our choice of Posterior Networks for our experiments was based on two considerations. First, we already had an existing implementation for one of our real-world applications and experiments using 2D-decision boundary algorithms yielded promising results. Second, for certain controlled scenarios (Fig 2), we could derive a closed analytic form for the model score estimation bias in terms of the model score, ratios of priors and observed evidence ($\gamma(x)$), which is positively correlated with the notion of “uncertainty” (entropy of the epistemic distribution) in Posterior networks, allowing us to compare empirical observations with expected behavior. We also report some results on MC-Dropout methods. (Sec 6.2, Pg 8, RQ2 and Appendix C.5, Table 5).
>
> **Comment 5 : Q2** The interpretation of $\gamma(x)$ in Section 3.2 is not really ... part may be further developed and discussed.
>
> **Response:**  Thanks for the suggestion. We mentioned the definition of $u(x) = H(q(x))$ in footnote 3 (Pg 2) but agree that further explanation would be helpful. We will expand on this item with a small plot of $u(x)$ and $\gamma(x)$ as well as explanation of the positive correlation either in the main paper or the appendix in the revised draft.
>
> **Comment 6 : Q3:** Can you provide insights on generalizing the analysis of estimation bias to other settings (as compared to Fig. 2) ?
>
> **Response:**  The generality of Figure 2 is primarily limited by the choices of (i) the global prior (Beta distribution) and (ii) estimation of model score and uncertainty using Posterior networks.
>
> Note that Thm 3.1(a) on the expectation of test/true conditional class probabilities conditioned on the train positive class probability holds true independent of assumption (ii). For any alternative method of estimating model score and uncertainty, we can derive the relationship between the model score (i.e., model estimated $p(y=1|x)$ ) and test positivity rate based on the dependency between the train positivity and the model score. The term ( $\lambda(x)$ ), which is related to the density of observations will continue to incorporate the effect of epsitemic uncertainty.
>
> Figuring out an equivalent closed form result for Thm 3.1(a) is non-trivial when assumption (i) does not hold. However, for a global unimodal prior (not necessarily a Beta distribution), the expectation of test/true conditional class probabilities conditioned on the train positive class probability is shifted away from the train distribution in the direction of the global prior since the posterior distribution of the true positivity conditioned o the train positivity combines the effects of the evidence (train positivity) and the global prior.

---

> ### Author Response · Authors · 2023-11-21
> **Response to Reviewer i837 [3/4]**
>
> **Comment 7: Q4:** The decision-making approach seems debatable.
> **Response:** Please see the comments below
>
> **Comment 7(a):** Q4.1 In Section 4: is it reasonable to leave $b$ unconstrained ? Shouldn't a monotonicity assumption be imposed ?
>
> **Response:** Since for each uncertainty level $i$, there is a single threshold $b_i$, we do assume monotonicity of the true conditional probability of the positive class with respect to the model score for a fixed uncertainty but not with respect to uncertainty for a fixed model score. This is because monotonicity with respect to uncertainty that does not hold true across the entire score range. Figure 3(a) shows the behavior (flipped trends for different score ranges) for the case of Beta priors with uncertainty estimated using Posterior Networks.
>
> In fact, we did experiment with a variant (EW_DPMT-MONO) of the DP algorithm where we constrained the decision boundary such that the true conditional probability is monotonically decreasing in uncertainty (i.e., $b_i \geq b_j$ for $i > j$, the boundary function to be look like a staircase). This variant has higher complexity $O(K^2L^3)$ than EW_DPMT where $K$ and $L$ are the number of uncertainty and score bins. The Table below shows a comparison of the performance of the two algorithms for the same bin configuration. We observe that for high precision bound (as in Criteo), the monotonicity assumption does hold, but that is not true as the precision bound is lowered and the unconstrained version results in better performance.
>
> Table 1: Performance with different unconstrained and monotonic variants of EW-DPMT with uncertainty bins = 100 and score bins = 3. Note that here we used a smaller number of uncertainty bins instead of 500 (in the paper) to reduce computational effort.
>
> |   |   |   |
> |---|---|---|
> |Dataset|Criteo (Tau=3) [Recall@90% Precision]|Avazu (Tau=5) [Recall@70% Precision]|
> ||Score_bins= 100, Unc_bins=3|Score_bins= 100, Unc_bins=3|
> |ST|2.2±0.2%|1.9±0.6%|
> |EW_DPMT|2.6±0.3%|2.3±0.6%|
> |EW_DPMT - MONO|2.6±0.3%|1.9±0.6%|
>
> **Comment 7(b):** Q4.2 Is it reasonable to always keep the decision-making process precise ? Wouldn't it be preferable to abstain in presence of a high (epistemic) uncertainty ?
>
> **Response:** In the setting we are considering, we are attempting to optimize recall of detecting something as positive (e.g., fraudulent transaction or malignant tumor) or not while maintaining a certain precision level. Here, we are not distinguishing between abstaining and a negative decision since we are focusing on the precision and recall of positive detection.  We could reformulate the problem to find two decision boundaries that separate out three regions: positives, abstain, and negatives (e.g., malignant tumor, needs further testing, no tumor) but need to redefine the objective function appropriately.
>
>
> **Comment 7(c):** Q4.3 Aleatoric and epistemic uncertainties are here not distinguished from each other, which seems to be an issue: what if u(x) is maximal ? Is it reasonable to make decisions in this case ?
>
> **Response:**  Making a decision about positive class prediction, abstaining or negative class prediction based only on uncertainty level of a bin is likely to be suboptimal. Instead, it is preferable to consider the model output score, uncertainty level as well as the actual positive class probability in the calibration set for the corresponding bin. If the actual positive class probability  is high, then the bin will get picked to be on the positive side of the decision boundary. (Similarly with negative class and abstaining if neither is high).
>
>
> Since aleatoric uncertainty is a deterministic function of the model estimated conditional probability (model score), the additional predictive power of the uncertainty signal primarily comes from the ‘epistemic uncertainty’. In case of the posterior networks, the differential entropy of the ‘epistemic distribution’ q(x) includes contributions from both aleatoric and epistemic uncertainty. However, this is not an issue since the calibration approach adjusts for it and utilises the non-redundant information given the model score.
>
> **Comment 7(d):** Q4.4 What is the effect of discretizing the Score-Uncertainty space ? To this extent, shouldn't the independent splitting on both dimensions avoided ?
>
> **Response:** Fig 5(a-b) show the dependence of performance on varying bin counts. In general, very fine bins lead to high computation and poor generalization, while coarse bins lead to over smoothing even if the compute costs are lower. There is a sweet spot that is dataset dependent.
>
> We did not understand the comment on avoiding the independent splitting. In Sec 6.2, Table 1, we do report results on the equi-span setting with independent splitting. In most cases, the resulting recall @precision bound from nested splitting is comparable or better.

---

> ### Author Response · Authors · 2023-11-21
> **Response to Reviewer i837 [4/4]**
>
> **Comment 8: Q5**  The "proof" that the partitioning problem of the Score-Uncertainty space is NP-hard (mentioned in the introduction) seems to directly proceed from the bin-packing problem behind.
>
> **Response:**  Please note that partitioning of the score-uncertainty space does not readily map to any of the standard bin-packing or knapsack problems because while the profit (recall) is additive, the cost (1-precision) is not additive and there is a dependence on the bins selected till that point. The NP-hardness result follows from a mapping to the subset-sum problem through a careful constructed reduction (Appendix F). To the best of our knowledge, this is not a previously published finding and there are potentially other applications for this result.
>
>
> **Comment 9: Q6:** Equi-weight binning case, "it is possible that ... can you be a bit more specific ?
>
> **Response:**  Thanks for the question. We will rewrite the line to explain it better. Let us say we wish to maximize recall for a precision bound of $\sigma$. Let the current selected bins have a net total of N samples with P of them positive. Given two bins A and B with total and positive samples and $(n_A, p_A)$ and $(n_B, p_B)$. Even if $p_A/n_A > p_B/n_B$, it might be preferable to choose bin B over bin A since it is possible that $(P+p_B)/(N +n_B) > \sigma > (P+p_A)/(N +n_A)$ when the bin sizes $(n_A, n_B)$ are different. For example, consider the case where P=10, N=20, p_A=5, n_A = 15, p_B = 1, and n_B =4 . Here $p_A/n_A = 5/15 > p_B/n_B = 1/4$, but $( P+p_B)/(N +n_B) = 11/24 > (P+p_A)/(N +n_A) = 15/35$.
>
> **Comment 10: Q7:** Binning strategy: I understand ... the bins computed.
>
> **Response:**  We explore three binning strategies (a) independent binning or equ—span (b) splitting first on uncertainty followed by score [Unc-Score] (c) splitting first on score followed by uncertainty [Score-Unc]. In Sec 6.1, Table 1, we present results on (a) and (b). We skipped including results on (c) for the lack of space and to avoid confusion. The table below summarizes the results for binning strategy (c) which yields similar results. There is also a dependence on the number of bins chosen (see Figures 5(a) and 5(b)) In future, it might able be preferable to explore perform binning and decision boundary detection jointly.
>
> Table 2: Performance with different equi-weight binning strategies. Score-Unc involves splitting on Score quantiles followed by that Uncertainty while it is the opposite for Unc-Score. Same number of score and uncertainty bins is used in both.
>
> | Dataset                | Criteo (Tau=3) [Recall@90% precision] |                | Avazu (Tau=5) [Recall@70% precision] |                |
> |------------------------|-------|----------------|-------|----------------|
> | Binning nest Order         | Score-Unc   | Unc-Score   | Score-Unc   | Unc-Score   |
> | #bins                  | (500, 3)    | (3, 500)    | (500, 3)    | (3, 500)    |
> | ST                     | 2.2±0.2%  | 2.2±0.2%  | 1.9±0.6%  | 1.9±0.6%  |
> | MIST                   | 2.6±0.2%  | 2.6±0.3%  | 2.9±0.2%  | 2.6±0.3%  |
> | GMT                    | 2.6±0.4%  | 2.7±0.3%  | 2.9±0.2%  | 2.7±0.3%  |
> | EW_DPMT                | 2.7±0.2%  | 2.7±0.3%  | 2.9±0.2%  | 2.7±0.3%  |

---

> > ### Comment · Reviewer_i837 · 2023-11-22
> > **Follow-up**
> >
> > I would like to thank the authors for their detailed answers to my comments and questions.
> >
> > I agree that these clarifications better enhance the contributions of the submission; I have raised my score accordingly.

---

### Official Review · Reviewer_ZEGe · 2023-10-31

**Soundness:** 3 good
**Presentation:** 3 good
**Contribution:** 3 good
**Rating:** 6
**Confidence:** 3

**Summary:**

Paper considers new approach to uncertainty estimation for calibrating the model and improving the classification accuracy in the case of (imbalanced) training set and/or dataset drift. Theoretical and empirical analysis of model output score dependence of uncertainty and the score are evaluated, and based on dynamic programming and isotonic regression an approximation algorithms are formulated for general NP-hard problem of decision boundary selections using both score and uncertainty. Three benchmark datasets, from online advertising and e-commerce domain, are utilised to shown the usefulness of the proposed approach, compared to related algorithms.

**Strengths:**

Paper proposes novel theoretical analysis of combining predictive uncertainty and classification score for decision boundary selection. The findings are supported by empirical analysis and versatile set of tests. It is shown that optimisation problem is NP-hard, and the new practical algorithms to approximate the solution with dynamic programming and isotonic regression based approaches, are presented. To my knowledge, the analysis and proposed algorithms for this particular problem are novel contributions, showing improvements against some related score only and 2D decision boundary selection algorithms on real-world benchmark datasets.

**Weaknesses:**

The proposed approach is motivated by the applications such as medical diagnosis and fraud detection. However, only advertising and e-commerce datasets are considered in the empirical experiment section. It would strengthen the work, if other additional imbalanced datasets from different application domains could have been experimented, as well.

**Questions:**

- Why choosing the particular datasets? (additional data from some other application domains could support the work and findings further)

---

> ### Author Response · Authors · 2023-11-20
> **Response to Reviewer ZEGe [1/1]**
>
> We thank the reviewer for their feedback and share our response below.
>
> **Comment 1** : Q1: The proposed approach is motivated by the applications such as medical diagnosis and fraud detection. However, only advertising and e-commerce datasets are considered in the empirical experiment section. It would strengthen the work, if other additional imbalanced datasets from different application domains could have been experimented, as well.
> Why choosing the particular datasets? (additional data from some other application domains could support the work and findings further)
>
>
> **Response** : Due to the space limitation we could only present results on a few datasets, but we do wish to clarify that the E-Com dataset was derived from a proprietary fraud/abuse detection application. Certain transformations were applied to avoid disclosing sensitive information while preserving the core integrity of the data for research purposes.

---

> > ### Comment · Reviewer_ZEGe · 2023-11-23
> > **Response to rebuttal**
> >
> > Thanks for the response. I have read all reviews and rebuttals, and authors have clarified many concerns. Manuscript shows promising results combining model score and uncertainty, although empirical datasets could have been more versatile. I keep my original score.

---

### Author Response · Authors · 2023-11-22
**Note To Area Chairs and All the Reviewers**

We thank all the reviewers for their detailed comments and suggestions. We have attempted to address the reviewer concerns and questions to our best including additional experimental results.

Below we summarize the key items of our response:

**Presentation issues**:  We acknowledge the various gaps in our presentation pointed out by Reviewers [i837] and [D6i1].

We will rephrase the contributions to further clarify our key contributions, the connection between (estimation bias analysis) Section 3 and 2D decision making, as well as the scope of our results. Additionally, we will make ensure any non-standard terms like “test positivity” that we used for brevity are defined precisely the first time they are introduced and enhance the notation table in the appendix. We will also fix the typos pointed out by the reviewers and do further proof reading of the revised version.  Please see Reviewer i837: Comments -2, 3  and  Reviewer D6i1: Comments- 1,2,3,6,7,8,9.

**Gaps in Literature review**:  Due to lack of space, we only included a limited discussion of related work [Section 2] in our submission.  In our official comments in response to the reviewers, we have clarified how our work is different from some of the previous works that deal with Decision making accounting for uncertainty and set-value predictions in a multi-class setting.  We also clarify the scope of our work and how recent work by Bengs et al.  on the drawbacks of Posterior network based uncertainty estimation is orthogonal to and does not invalidate our contributions. We will do our best to incorporate these explanations into the revised related work with an additional section in the appendix if necessary.  Please see Reviewer i837: Comments -1,4  and  Reviewer D6i1: Comments-4 for more details.


**Choice of posterior networks**:  Reviewers [i837] and [D6i1] raised concerns  on our choice of Posterior networks for epistemic uncertainty quantification, given the recent work of Bengs et al. on Pitfalls of epistemic uncertainty quantification through loss minimization, Neurips 2022.  We understand the concern and and have attempted to explain that our primary results  on the estimation bias , i.e., deviation between conditional probability of the positive class in the training set and the associated expectation of the conditional probability in the test set [ Theorem 3.1(a)] and the 2-D decision-boundary estimation are independent of the choice of Posterior networks. While we do provide the relationship between model estimate and the test data observation of the conditional probability of the positive class $p(Y=1|x)$   for Posterior networks in Theorem 3.1(b), the intent was only to facilitate comparison of theoretical and empirical results.  We do not  qualify the goodness of any particular technique for modeling epistemic uncertainty. We  expect that it is quite likely that other approaches beyond Posterior Networks could yield superior results in terms of optimizing decision boundaries and this remains an area of further exploration. We, in fact, provide results on MC-Dropout methods  as well (Sec 6.2, Pg 8, RQ2 and Appendix C.5, Table 5) and will attempt to include results from other recent approaches in the Appendix.  Please see Reviewer i837: Comments -4  and  Reviewer D6i1: Comments-4 for more details.

**Decision boundary estimation**: Reviewer [i837] had multiple thoughtful questions on our decision boundary estimation. We have tried our best to address these questions in our response and will  incorporate these explanations on different binning strategies, monotonicity assumptions, and partitioning into three regions (positive, abstain, negative) in our revision. Please see Reviewer i837: Comments - 7(a-d), 8, 9,10.


Thanks again for the review process and your valuable comments,  which should help us improve the clarity of our submission. If there are any comments/questions we overlooked or if there are further ways to improve the paper, please let us know and we will be glad to work on them.

---

### Meta-Review · Area_Chair_ckyo · 2023-12-12

**Metareview:**

The paper furthers the science of exploiting uncertainty for classification problems. There is consensus that the paper has merit. But it is not well presented. It is thus difficult to make a strong case for acceptance.

**Justification For Why Not Higher Score:**

There are presentation issues.

**Justification For Why Not Lower Score:**

It makes several contributions which should be of interest to ICLR community

---

### Decision · Program_Chairs · 2024-01-16

Accept (poster)